# Cell Line-Dependent Internalization, Persistence, and Immunomodulatory Effects of *Staphylococcus aureus* in Triple-Negative Breast Cancer

**DOI:** 10.3390/cancers17182947

**Published:** 2025-09-09

**Authors:** Sima Kianpour Rad, Runhao Li, Kenny K. L. Yeo, Clare Cooksley, Gohar Shaghayegh, Sarah Vreugde, Fangmeinuo Wu, Yoko Tomita, Timothy J. Price, Wendy V. Ingman, Amanda R. Townsend, Eric Smith

**Affiliations:** 1Solid Tumour Group, Basil Hetzel Institute for Translational Health Research, The Queen Elizabeth Hospital, Central Adelaide Local Health Network, Woodville South, Adelaide, SA 5011, Australia; sima.kianpourrad@adelaide.edu.au (S.K.R.); runhao.li@adelaide.edu.au (R.L.); kenny.yeo@adelaide.edu.au (K.K.L.Y.); fangmeinuo.wu@adelaide.edu.au (F.W.); yoko.tomita@sa.gov.au (Y.T.); timothy.price@sa.gov.au (T.J.P.); amanda.townsend@sa.gov.au (A.R.T.); 2Adelaide Medical School, The University of Adelaide, Adelaide, SA 5005, Australia; clare.cooksley@adelaide.edu.au (C.C.); gohar.shaghayegh@adelaide.edu.au (G.S.); sarah.vreugde@adelaide.edu.au (S.V.); wendy.ingman@adelaide.edu.au (W.V.I.); 3Department of Surgery-Otolaryngology Head and Neck Surgery, The University of Adelaide and Basil Hetzel Institute for Translational Health Research, Central Adelaide Local Health Network, Woodville South, SA 5011, Australia; 4Discipline of Surgery, The University of Adelaide, Adelaide, SA 5005, Australia; 5Medical Oncology, The Queen Elizabeth Hospital, Central Adelaide Local Health Network, Woodville South, Adelaide, SA 5011, Australia; 6Robinson Research Institute, The University of Adelaide, Adelaide, SA 5005, Australia

**Keywords:** triple-negative breast cancer (TNBC), *Staphylococcus aureus*, intracellular bacteria, PD-L1 expression, Toll-like receptor 2 (TLR2), STAT1 signaling, immune checkpoint, bacterial persistence, immunotherapy

## Abstract

Triple-negative breast cancer (TNBC) is an aggressive form of breast cancer with few effective treatment options. Immunotherapy, which boosts the immune system’s ability to fight cancer, has shown promise, but many patients do not respond well. Recent studies suggest that bacteria found inside tumors might affect cancer growth and treatment success. In this study, we examined whether *Staphylococcus aureus*—a common bacterium present in breast tissue—can survive inside TNBC cells and influence how they interact with the immune system. We found that *S. aureus* increases the levels of a protein called PD-L1, which cancer cells use to hide from immune attack. This finding suggests that bacteria may help cancer cells resist immunotherapy. Understanding these bacteria–cancer cell interactions could lead to new treatments that target both cancer and bacteria to improve outcomes for patients.

## 1. Introduction

Triple-negative breast cancer (TNBC) is an aggressive subtype defined by the absence of estrogen receptor, progesterone receptor, and human epidermal growth factor receptor 2 expression. Lacking targeted therapies, TNBC is associated with high recurrence rates and poor prognosis [1]. Molecular subtyping has further classified TNBC into biologically distinct groups, including basal-like (BL1 and BL2), mesenchymal (including mesenchymal stem-like and claudin-low), and luminal androgen receptor subtypes [2,3]—each differing in proliferation rates, immune infiltration, and drug responsiveness.

Immunotherapy targeting the PD-1/PD-L1 immune checkpoint pathway has emerged as a promising approach for TNBC, particularly in patients with high PD-L1 expression [4,5]. PD-1, an immune checkpoint receptor on T-cells, interacts with its ligands PD-L1 and PD-L2 to suppress immune responses, enabling tumor immune evasion [6,7]. Immune checkpoint inhibitors (ICIs) block this interaction, restoring T-cell function and enhancing antitumor immunity [8]. PD-L1 expression is especially elevated in basal-like and metastatic TNBC tumors [9], making ICIs such as pembrolizumab viable therapeutic options. However, response rates remain inconsistent, and resistance mechanisms—such as dysregulation of the interferon-gamma (IFN-γ)/JAK-STAT pathway—continue to limit clinical benefit [10,11]. Identifying alternative regulatory mechanisms of PD-L1 expression is therefore critical to improving outcomes in TNBC.

One emerging area of interest is the role of bacteria within tumors. While breast tissue was once considered sterile, recent studies have demonstrated the presence of bacteria in both normal and malignant breast tissue [6,12,13,14,15,16,17,18,19,20,21,22]. Our recent meta-analysis of 11 studies using 16S rRNA sequencing on 1260 fresh breast tissue samples identified *Staphylococcus* as one of the most frequently detected genera [23]. Further analysis of TCGA-BRCA RNA-sequencing data revealed that high *Staphylococcus* abundance correlated with proliferation-related gene expression and a 4.1-fold increased risk of mortality [23]. Although intratumoral bacteria are well studied in high-biomass cancers such as colorectal and oral cancer [24,25], their functional relevance in breast cancer—a traditionally low-biomass tumor—remains poorly understood.

Microbes may enter tumors via hematogenous spread, translocation from mucosal sites, infiltration of infected immune cells, or direct migration from adjacent tissues such as the skin or breast glands, although these routes are not fully defined. Once inside, bacteria can influence tumor biology, with animal studies showing that various intracellular species promote metastasis in breast cancer models [6,26,27]. In parallel, bacterial components and engineered microbes are being investigated for their ability to stimulate anti-tumor immunity or serve as vectors for drug delivery in cancer therapy [28,29,30,31], highlighting the dual potential of intratumoral bacteria as both drivers and disruptors of tumor progression.

*Staphylococcus aureus*, a facultative intracellular pathogen, is capable of surviving within both epithelial and immune cells, thereby evading immune surveillance and antibiotic treatment [32,33,34]. Though traditionally regarded as extracellular, *Staphylococcus* has been detected inside cancerous and immune cells in tumors [18], raising new questions about its functional role in cancer. *S. aureus* produces numerous virulence factors—including toxins and cell wall components—that modulate host cell signaling, proliferation, and immune responses [35]. Notably, *S. aureus* can activate Toll-like receptor 2 (TLR2) signaling and has been shown to induce PD-L1 expression in immune cells [36]. In head and neck squamous cell carcinoma models, TLR2 agonist including heat-killed *S. aureus* and synthetic Pam3CSK4 increase PD-L1 expression [37]. More recently, a murine TNBC study demonstrated that treatment with *S. aureus* culture supernatant or α-hemolysin—but not heat-killed bacteria—enhanced CD8+ T-cell infiltration, increased tumor PD-L1 expression, and improved response to immune checkpoint blockade [38].

This knowledge gap underscores the need to investigate whether *S. aureus* can persist in TNBC cells and alter immune checkpoint regulation.

To address this, we employed a panel of well-characterized breast cell lines representative of key TNBC subtypes: MDA-MB-468 (basal-like 1), MDA-MB-231 and Hs578T (mesenchymal stem-like), BT-549 and CAL-51 (mesenchymal), and MDA-MB-453 (luminal androgen receptor), alongside the non-tumorigenic mammary epithelial line MCF-12A. These models differ in baseline PD-L1 expression, epithelial–mesenchymal phenotype, and innate immune signaling capacity, providing a valuable system to dissect host–microbe interactions.

In this study, we aimed to determine whether viable *S. aureus* can invade and persist within human TNBC cells and to assess its functional consequences. We evaluated bacterial uptake, persistence, and cytotoxicity across this cell line panel and further examined whether intracellular *S. aureus* modulates PD-L1 expression through TLR2/STAT1 signaling in the context of IFN-γ stimulation. Together, these experiments were designed to provide mechanistic insight into how intratumoral bacteria may influence immune evasion and responsiveness to immune checkpoint blockade in TNBC.

## 2. Materials and Methods

### 2.1. Reagents

Interferon gamma (IFN-γ) was purchased from BioLegend (San Diego, CA, USA, Cat# 570204). Purified *S. aureus* lipoteichoic acid (LTA) (Cat# tlrl-pslta) and the synthetic TLR2 agonist Pam3CSK4 (Cat# tlrl-pms) were obtained from InvivoGen (San Diego, CA, USA).

### 2.2. Breast Cell Lines

TNBC cell lines—including MDA-MB-468 (ATCC; Cat# HTB-132), (ATCC; Cat# HTB-26), and MDA-MB-453 (ATCC; Cat# HTB-131), Hs578T (ATCC; Cat# HTB-126), BT-549 (ATCC; Cat# HTB-122), as well as the non-tumorigenic epithelial cell line MCF-12A (ATCC; Cat# CRL-10782), were obtained from American Type Culture Collection (ATCC, Manassas, VA, USA). The CAL-51 cell line (DSMZ, Cat# ACC 302) was obtained from the Leibniz Institute DSMZ—German Collection of Microorganisms and Cell Cultures GmbH (Braunschweig, Germany). Cell line identity was authenticated by short tandem repeat (STR) profiling, and all lines were confirmed to be free of mycoplasma contamination using PCR-based testing [39].

MDA-MB-231, MDA-MB-468, MDA-MB-453, Hs578T, and CAL-51 were maintained in Dulbecco’s Modified Eagle Medium (DMEM; Gibco, Thermo Fisher Scientific, Waltham, MA, USA; Cat# 12430062), and BT-549 cells were cultured in RPMI Medium 1640 (Gibco; Thermo Fisher Scientific, Cat#11875-093). Both media formulations contain 4.5 mg/mL glucose. All media were supplemented with 10% fetal bovine serum (FBS; Gibco, Thermo Fisher Scientific; Cat# 26140079) and 1% penicillin-streptomycin (Gibco, Thermo Fisher Scientific; Cat# 15140122). MCF-12A cells were maintained in a 1:1 mixture of DMEM and Ham’s F12 (Gibco, Thermo Fisher Scientific; Cat# 11765054) supplemented with 20 ng/mL human epidermal growth factor (Sigma-Aldrich, St. Louis, MO, USA; Cat# 01-107), 100 ng/mL cholera toxin (Sigma-Aldrich; Cat# C8052), 0.01 mg/mL bovine insulin (Sigma-Aldrich; Cat# 16634), 500 ng/mL hydrocortisone (Sigma-Aldrich; Cat# H0888), 5% heat-inactivated horse serum (Sigma-Aldrich; Cat# H1138) and 1% penicillin-streptomycin.

These cell lines were selected to represent the major molecular subtypes of TNBC (basal-like, mesenchymal/mesenchymal stem-like, and luminal androgen receptor), which differ in PD-L1 expression, epithelial–mesenchymal phenotype, and innate immune signaling capacity. This diversity provided a comprehensive model system to investigate host–microbe interactions.

### 2.3. Labeling of S. aureus with eFluor450

The *Staphylococcus aureus* ATCC 25923 reference strain was initially cultured on Tryptone Soya Agar (TSA) (Oxoid, Thermo Fisher Scientific; Cat# CM0131). Colonies were then inoculated into 35 mL of Tryptone Soya Broth (TSB) (Oxoid, Thermo Fisher Scientific; Cat# CM0129) in 50 mL tubes and incubated overnight at 37 °C under aerobic conditions with agitation (180 rpm). For fluorescent labeling, the overnight culture was adjusted to a concentration of 1.0 × 10^9^ colony-forming units per milliliter (CFU/mL), pelleted by centrifugation at 3200× *g* for 10 min, and washed three times with sterile, protein-free phosphate-buffered saline (PBS). The bacterial pellet was resuspended in 750 µL of 10 µM eBioscience Cell Proliferation Dye eFluor450 (Invitrogen, Thermo Fisher Scientific, San Diego, CA, USA; Cat# 65-0842-90) (according to the protocol provided with the kit) and incubated at 37 °C in the dark for 30 min with gentle agitation. Excess dye was quenched by incubating the labeled bacteria in DMEM containing 10% FBS for 10 min, followed by two additional PBS washes. eFluor450 was selected for its bright and stable signal in the violet channel, which allows sensitive detection of bacteria by flow cytometry while minimizing spectral overlap and host cell autofluorescence.

For infection, a bacterial suspension was prepared at McFarland standard 0.5, corresponding to approximately 1.5 × 10^8^ CFU/mL, and diluted in DMEM to achieve the desired multiplicity of infection (MOI) before being added to the cells.

### 2.4. Gentamicin Protection Assay

Breast cell lines were seeded at 1.5 × 10^5^ cells per well in 12-well plates and incubated overnight at 37 °C in 5% CO_2_. The following day, cells were washed with Dulbecco’s Phosphate-Buffered Saline (DPBS) (Gibco, Thermo Fisher Scientific; Cat# 14190144) and infected with eFluor450-labeled or unlabeled *S. aureus* in DMEM supplemented with 10% FBS, without antibiotics, at MOIs of 10, 50, or 200 for 2 h at 37 °C in 5% CO_2_. To minimize photobleaching, infections were performed in the dark.

Following infection, cells were washed three times with sterile DPBS, incubated in DMEM supplemented with 10% FBS and 200 µg/mL gentamicin (Gibco, Thermo Fisher Scientific; Cat# 15750060200) for 1 h, followed by two washes with DPBS to eliminate extracellular bacteria. Cells were then incubated in their respective maintenance medium supplemented with either 50 µg/mL gentamicin for 24 h experiments, or 5 µg/mL gentamicin for experiments longer than 24 h.

To verify the efficacy of gentamicin in eliminating extracellular bacteria, a minimum inhibitory concentration (MIC) assay was conducted following established protocols [40].

### 2.5. Measurement of S. aureus Internalization by Flow Cytometry

Breast cell lines were infected with unlabeled or eFluor450-labeled *S. aureus* at MOIs of 10, 50 and 200 bacteria per cell. At 24 or 48 h post-infection, the cells were detached using TrypLE Express Enzyme without phenol red (Gibco, Thermo Fisher Scientific; Cat# 12604021), centrifuged at 300× *g* for 5 min, and washed twice with DPBS.

To distinguish viable from dead cells, 100 µL of ViaDye Red Fixable Viability Dye (Cytek Biosciences, Fremont, CA, USA, Cat# R7-60008) at a dilution of 1:100,000 in protein-free PBS was added. Cells were incubated in the dark for 20 min, followed by two washes with DPBS.

A minimum of 30,000 single-cell events per sample were acquired using a Cytek Aurora spectral flow cytometer (Cytek Biosciences). Cells that were positive for ViaDye Red Fixable Viability Dye were excluded from analysis. The percentage *S. aureus*-positive cells were determined by comparing eFluor450 fluorescence intensity to that of cells infected with unlabeled bacteria. The intracellular bacterial load was quantified as the eFluor450 geometric mean fluorescence intensity (GMFI). All analyses were preformed using FlowJo v10.10.0 (BD Biosciences, San Jose, CA, USA).

### 2.6. Cytotoxicity and Inhibition of Proliferation

Breast cells were infected with *S. aureus* at MOIs of 10, 50 and 200 for 2 h. Uninfected cells served as controls. Following infection, cells were washed twice with sterile DPBS and incubated in DMEM supplemented with 10% FBS and 200 µg/mL gentamicin for 1 h, followed by 2 washes with DPBS and incubation with gentamycin 50 µg/mL for 2 h. Cells were washed twice with DPBS, detached using TrypLE and counted. Cell counts were normalized to uninfected cells.

To assess the effect of *S. aureus* infection on cell proliferation, infected cells were seeded at a density of 3.3 × 10^3^ cells per well in 96-well plates (Corning Incorporated, Corning, NY, USA; Cat# 3696). Cells were then incubated in the respective maintenance media supplemented with 5 µg/mL gentamicin for 5 days and cell proliferation was assessed using a crystal violet staining assay, as previously described [41]. Crystal violet staining was used to quantify adherent host cell biomass as a proxy for viability and proliferation. Although the dye can also stain bacteria, the contribution of internalized organisms is negligible compared with host cell content due to prior gentamicin treatment and the low bacterial load. Thus, the absorbance measurement primarily reflects mammalian cell proliferation. Proliferation of infected cells was compared to the uninfected controls.

*S. aureus* exoproteins were prepared as previously described [42]. Briefly, conditioned TSB from 15 h *S. aureus* cultures was centrifuged at 4000× *g* for 10 min at 4 °C to remove bacterial cells and filtered through a 0.22 μm Acrodisc syringe filter (Pall Corporation, Port Washington, NY, USA; Cat# 4525). Exoproteins were then concentrated using Pierce Protein Concentrators PES with a 3 kDa molecular weight cutoff (Thermo Fisher Scientific; Cat# 88512), and total protein content was quantified using the Pierce BCA Protein Assay Kit (Thermo Fisher Scientific; Cat# 23225). Serial dilutions of the exoproteins were prepared in the respective maintenance media supplemented with 5 µg/mL gentamicin. Cells were seeded at a density of 3.3 × 10^3^ cells per well in 96-well plates. The following day, the culture media was replaced with *S. aureus* exoproteins, and cells were treated for 5 days. Cell proliferation was assessed using a crystal violet staining assay, as previously described [41]. Proliferation in exoprotein treated cells was compared to untreated controls.

### 2.7. Expression of Cell Surface Markers Determined by Flow Cytometry

Cells were infected with various MOIs of unlabeled *S. aureus* as described previously. Following infection, both infected and uninfected control cells were treated with IFN-γ and incubated for 24 or 48 h. In parallel, cells were also treated with TLR2 agonists for 24 h, with or without IFN-γ, to evaluate the impact of TLR2 activation on cellular responses. As part of the control conditions, a subset of uninfected cells was maintained with and without IFN-γ treatment. The surface expression levels of PD-L1 and TLR2 were then measured by flow cytometry.

At the designated time points, 3 × 10^5^ cells from each group were harvested, washed twice with PBS, and stained with 100 µL of ViaDye Red Fixable Viability Dye (1:10,000 dilution). After staining, cells were washed twice with PBS and resuspended in ice-cold FACS buffer (1% FBS and 0.05% sodium azide in PBS).

To reduce non-specific antibody binding, cells were incubated with Human BD Fc Block (1:50 dilution; BD Pharmingen, San Jose, CA, USA; Cat# 564220) for 10 min at room temperature. Cells were then washed with FACS buffer prior to antibody staining.

Cells were stained in a final volume of 60 µL with the following antibodies, which were titrated before use to determine optimal dilution: PD-L1, Brilliant Violet 421 anti-human CD274 (1:20 dilution; BioLegend, San Diego, CA, USA; Cat# 329714); TLR2, PE/Cyanine7 anti-human CD282 Antibody (1:20 dilution; BioLegend; Cat# 309722). Cells were incubated in the dark at 4 °C for 30 min, followed by fixation with 4% paraformaldehyde for 10 min. Cells were then washed twice with FACS buffer before flow cytometry analysis.

Flow cytometry was performed using the Cytek Aurora spectral flow cytometer (Cytek Biosciences, Fremont, CA, USA), acquiring a minimum of 30,000 events per sample. Autofluorescence was controlled by including unstained cell controls containing unlabeled bacteria. Heat killed cells (60 °C for 5 min) containing unlabeled bacteria served as a positive control for the live/dead Viability Dye. Single-stained controls (cells with unlabeled bacteria) were used for compensation and background correction. Marker expression was quantified using the GMFI. All analyses were performed using FlowJo v10.10.0.

### 2.8. Western Blot

MDA-MB-468 and MDA-MB-231 were infected with *S. aureus* at MOI 50, while MDA-MB-453 and MCF-12A were infected at MOI 200, to maximize infection efficiency while maintaining high cell viability. After 24 h, the cells were washed three times with cold DPBS and lysed using RIPA Lysis and Extraction Buffer (Thermo Fisher Scientific; Cat# 89900) supplemented with Halt Protease Inhibitor Cocktail (Thermo Fisher Scientific; Cat# 78438) and Halt Phosphatase Inhibitor Cocktail (Thermo Fisher Scientific; Cat# 78420). Lysates were homogenized by passage through a 26-gauge needle, and insoluble debris was removed by centrifugation at 17,000× *g* for 15 min at 4 °C. Total protein concentrations were determined using Pierce BCA Protein Assay Kit. Protein amounts of 30 µg and 40 µg were used for p-STAT1 and STAT1 detection, respectively. Proteins were separated by SDS-PAGE on 4–15% Mini-PROTEAN TGX Stain-Free Protein Gels (Bio-Rad Laboratories, Hercules, CA, USA; Cat# 4568084) and transferred onto 0.2 µm polyvinylidene difluoride (PVDF) membranes (Bio-Rad Cat# 1704156) using the Trans-Blot Turbo Transfer System (Bio-Rad). Membranes were blocked with 5% skim milk for at least 1 h then incubated overnight at 4 °C with primary antibodies at 1:1000 dilution in TBST with 0.1% skim milk powder. Primary antibodies were STAT1 monoclonal antibody (Thermo Fisher Scientific; Cat# MA5-15129, clone C.146.9), phospho-STAT1 (Tyr701) monoclonal antibody (Thermo Fisher Scientific; Cat# 33-3400, clone ST1P-11A5). After three washes with TBST, membranes were incubated for 1 h with 1:1000 goat anti-mouse (H+L)-HRP conjugated secondary antibody (Bio-Rad; Cat# 1721011) and 1:10,000 Precision Protein StrepTactin-HRP conjugate (Bio-Rad; Cat# 1610380) in TBST supplemented with 1% skim milk powder. Detection was performed using Clarity Western ECL Blotting Substrate (Bio-Rad; Cat# 1705060) on a ChemiDoc Gel Imaging System (Bio-Rad). Image Lab Software v6.0.1 (Bio-Rad) was used for band quantification, with β-actin as the loading control for normalization [39].

### 2.9. Clearance of Viable Intracellular S. aureus

To assess bacterial clearance, breast cell lines infected with *S. aureus* at MOIs of 50 and 200 were harvested daily for 7 days. Cells (3 × 10^4^) were lysed with 200 µL of 1% Triton X-100 for 5 min. The lysate was serially diluted in sterile PBS, plated onto TSA and incubated at 37 °C for 24 h. After incubation, the number of CFUs were counted, and bacterial concentration was calculated as CFU/mL using the formula: CFU/mL = (number of colonies × dilution factor)/volume plated (mL). At each time point, uninfected lysed cells were used as negative controls.

### 2.10. Transmission Electron Microscopy

Transmission electron microscopy (TEM) was used to confirm the presence and morphology of intracellular *S. aureus* within MDA-MB-231 cells. Cells were infected with a MOI of 200 and collected at 24 h and 7 days post-infection. A total of 5 × 10^5^ cells were fixed in 800 µL of 4% paraformaldehyde and 1.25% glutaraldehyde in PBS with 4% sucrose, pH 7.2 for at least 24 h at 4 °C. Following fixation, cells were washed twice with PBS containing 4% sucrose for 10 min each, and then post-fixed in 2% osmium tetroxide for 1 h on a rotator. Dehydration was performed using a graded ethanol series (70%, 95% and 100% ethanol, three changes of 20 min each), followed by 30 min in propylene oxide.

For resin infiltration, cells were incubated in a 1:1 mixture of propylene oxide and resin for 1 h, followed by overnight incubation in 100% resin, and an additional 4 h change in fresh 100% resin. Cells were then embedded in fresh resin and polymerized at 60 °C for at least 48 h.

After processing, 1 µm sections were cut using an ultramicrotome, stained with toluidine blue, and examined by brightfield microscopy to identify suitable regions for TEM imaging. Ultrathin sections (70 nm) were then cut, mounted on copper grids, and stained with 4% uranyl acetate and lead citrate. Imaging was performed using a Tecnai G2 Spirit 120 kV TEM, and images acquired using an AMT digital camera with v7.0.1 software.

### 2.11. Statistical Analysis

Statistical analyses were performed using Prism 10 for macOS (v10.4.0 (527), 23 October 2024; GraphPad Software Inc., La Jolla, CA, USA).

## 3. Results

### 3.1. Breast Cell Lines Exhibit Differential Internalization of Staphylococcus aureus

We assessed the capacity of *Staphylococcus aureus* (ATCC 25923) to internalize into a panel of triple-negative breast cancer (TNBC) cell lines representing the most prevalent molecular subtypes, as classified by Lehmann et al. (2011): MDA-MB-468 (basal-like 1), MDA-MB-231 and Hs578T (mesenchymal stem-like), BT-549 and CAL-51 (mesenchymal) and MDA-MB-453 (luminal androgen receptor) [2]. The non-tumorigenic MCF-12A breast epithelial cell line was included as a physiological comparator.

To distinguish intracellular from extracellular bacteria, we employed an optimized gentamicin protection assay (Appendix A). *S. aureus* was labeled with eFluor450, enabling flow cytometry and fluorescence microscopy for bacterial tracking. The eFluor450 labeling did not affect bacterial viability (Appendix A), and bacterial sensitivity to antibiotics was confirmed via a minimum inhibitory concentration (MIC) assay (Appendix A). The elimination of viable extracellular bacteria was validated using propidium iodide staining, which selectively labeled dead extracellular bacteria (Appendix A). Transmission electron microscopy (TEM) further confirmed the intracellular localization of *S. aureus* (Appendix A). The presence of viable intracellular bacteria was verified by plating lysed cell contents on agar 24 h post-infection, which resulted in colony formation (Appendix A). No colonies were detected in the culture media, confirming elimination of viable extracellular bacteria.

To evaluate internalization efficiency, breast cell lines were infected with eFluor450-labeled *S. aureus* at multiplicities of infection (MOI) of 10, 50, and 200, and analyzed by spectral flow cytometry 24 h post-infection (Figure 1 and Appendix A). The proportion of viable breast cells harboring intracellular bacteria was quantified via flow cytometry 24 h post-infection. In addition to infection rates, geometric mean fluorescence intensity (GMFI) provided an estimate of intracellular bacterial load per infected cell. Importantly, unlabeled *S. aureus* at comparable MOI did not significantly alter cellular autofluorescence (Appendix A), confirming that eFluor450 signal reflected true bacterial labeling.

Internalization increased in an MOI-dependent manner across all breast cell lines (Figure 1a,b; Appendix A). At MOI 10, an average of 67% of MDA-MB-468 and 54% of MDA-MB-231 cells were infected. Hs578T and BT-549 exhibited moderate internalization rates of 27% and 24%, respectively. In contrast, MDA-MB-453, CAL-51 and MCF-12A demonstrated minimal bacterial uptake, with internalization rates of 0.5%, 5%, and 9%, respectively. At MOI 50 and 200, nearly all MDA-MB-468 and MDA-MB-231 cells were infected (>90%). Internalization rates were moderate in Hs578T (48% at MOI 50 and 80% at MOI 200), BT-549 (36% at MOI 50 and 67% at MOI 200) and MCF-12A (33% at MOI 50 and 56% at MOI 200). In contrast, MDA-MB-453 (4% at MOI 50 and 27% at MOI 200) and CAL-51 (7% at MOI 50 and 14% at MOI 200) exhibited substantially lower internalization.

GMFI analysis showed that MDA-MB-468 consistently harbored a significantly higher intracellular bacterial load per cell than MDA-MB-231 (*p* < 0.0001 for all MOI; Figure 1c). BT-549 and Hs578T had moderate bacterial burdens, while MDA-MB-453, CAL-51 and MCF-12A exhibited the lowest intracellular bacterial burdens. The percentage of infected cells correlated with GMFI, highlighting a dose-dependent relationship between *S. aureus* internalization and intracellular bacterial burden (Figure 1d).

These findings reveal marked cell line–specific differences in susceptibility to *S. aureus* internalization and intracellular bacterial burden, highlighting the heterogeneity of bacterial–host interactions among breast epithelial models.

### 3.2. S. aureus Induces Dose-Dependent Cytotoxicity and Inhibits Proliferation in a Cell Line–Specific Manner

We next evaluated the acute (2 h) and long-term (5-day) effects of *S. aureus* infection on cell viability and proliferation using manual cell counts of viable cells and crystal violet assays, respectively (Figure 2a,b). The crystal violet assay, which quantifies adherent cell biomass as a surrogate for cumulative cell proliferation [41], was selected for its suitability in monitoring long-term growth dynamics following bacterial challenge.

At 2 h post-infection (Figure 2a), infection at MOI 200 significantly reduced viable cell numbers across all cell lines (*p* < 0.01), except CAL-51. MDA-MB-468 was the most susceptible, showing an 82% reduction in viable cells, followed by MDA-MB-231 (48% reduction). Hs578T and BT-549 were more moderately affected, with reductions of 30% and 27%, respectively. In contrast, MDA-MB-453, CAL-51 and MCF-12A showed minimal sensitivity with reductions of 13%, 0% and 16%, respectively. These cytotoxicity patterns correlated with bacterial internalization across cell lines (Spearman r = −1.000, *p* = 0.0004, *n* = 7; Appendix A), suggesting that intracellular bacterial load may underlie differential sensitivity. At MOI 50, only MDA-MB-468 and MDA-MB-231 exhibited a significant decline in cell numbers (*p* < 0.0001), while the other lines remained unaffected, indicating resistance to moderate bacterial loads. No significant reduction in viable cell numbers was observed in any cell lines at MOI 10.

To assess the long-term impact of *S. aureus* infection, we monitored cell proliferation over five days (Figure 2b). At MOI 10 and 50, all cell lines exhibited proliferation rates comparable to uninfected controls, suggesting tolerance to lower bacterial burdens. However, at MOI 200, proliferation was differentially affected. While BT-549, Hs578T, CAL-51, MDA-MB-453 and MCF-12A maintained their growth rates, MDA-MB-231 showed an approximate 20% reduction (*p* < 0.0001), and MDA-MB-468 exhibited a striking 74% decrease (*p* < 0.0001), consistent with its higher susceptibility to bacterial-induced cytotoxicity.

To further investigate the inhibitory effects on proliferation, we assessed whether *S. aureus* exotoxins (Figure 2c) or purified lipoteichoic acid (LTA) (Appendix A) contributed to growth suppression in selected cell lines. The half-maximal inhibitory concentration (IC50) of *S. aureus* exotoxin varied widely among cell lines, highlighting differential sensitivity. The IC50 values were 3.3 μg/mL (95% CI: 3–3.5 μg/mL) for MDA-MB-468, 10.0 μg/mL (95% CI: 8.7–11.4 μg/mL) for MDA-MB-231, 33.2 μg/mL (95% CI:31–35.4 μg/mL) for MDA-MB-453, and 54.5 μg/mL (95% CI: 51.4–57.4 μg/mL) for MCF-12A. In contrast, purified *S. aureus* LTA (1–50 µg/mL) had no significant effect on cell proliferation across all tested cell lines (Appendix A).

Altogether, the patterns of acute cytotoxicity and long-term growth inhibition correlated with the degree of bacterial internalization and intracellular bacterial burden, suggesting that susceptibility to *S. aureus*-induced cytotoxicity may be linked to intracellular bacterial load.

### 3.3. S. aureus Infection Enhances IFN-γ-Induced PD-L1 Expression in a Cell Line-Dependent Manner

PD-L1 is a critical immune checkpoint regulator and a therapeutic target in TNBC. While IFN-γ is a canonical inducer of PD-L1, previous studies show that bacterial products, including heat-killed *S. aureus*, can stimulate IFN-γ-producing lymphocytes [43,44,45,46,47]., and that microbial components can upregulate tumor PD-L1 in vivo [48,49,50,51,52]. Motivated by these observations, we investigated whether infection with live *S. aureus* modulates PD-L1 expression in breast cells, either directly or by altering responsiveness to IFN-γ.

Baseline surface PD-L1 expression across the panel of breast cell lines was first quantified using flow cytometry (Figure 3a). PD-L1 expression were highest in MDA-MB-231 and BT-549, with markedly lower levels in MCF-12A, Hs578T, MDA-MB-468, and MDA-MB-453. CAL-51 exhibited no detectable PD-L1.

Next, we assessed PD-L1 induction following IFN-γ stimulation in selected cell lines (Figure 3b,c), with the corresponding gating strategy shown in Appendix A. Because unlabeled *S. aureus* contributed to background signal and altered the PD-L1 histogram profile in spectral flow cytometry (Appendix A), all reference groups included cells infected with unlabeled *S. aureus* to control for spectral overlap and nonspecific background. The resulting histograms are presented in Appendix A. As expected, IFN-γ stimulation significantly increased PD-L1 expression at both 24 or 48 h (*p* < 0.0001). The degree of upregulation was inversely correlated with baseline expression. MDA-MB-468, which had low baseline PD-L1, showed the strongest response with an 11-fold increase at 24 h and 17-fold at 48 h. MCF-12A exhibited a 5-fold increase at 24 h, while MDA-MB-453 showed a 4.3-fold increase at 24 h and 4.6-fold increase at 48 h. In contrast, MDA-MB-231 cells showed only a modest 1.4-fold increase at 24 h and a 1.6-fold increase at 48 h, consistent with its already high baseline expression.

To determine whether *S. aureus* infection alone could induce PD-L1, we infected selected cell lines for 2 h and measured PD-L1 levels at 24 and 48 h post-infection. Across all lines, *S. aureus* infection alone did not significantly alter PD-L1 expression, regardless of MOI or time point.

However, co-treatment with IFN-γ and *S. aureus* revealed distinct cell line–dependent effects (Figure 3c, Appendix A). In MDA-MB-468, *S. aureus* infection significantly enhanced IFN-γ–induced PD-L1 expression, with a 1.9-fold (*p* < 0.0001) increase at 24 h and a 2.9-fold (*p* < 0.0001) increase at 48 h relative to IFN-γ-only controls. Similarly, in MDA-MB-231, *S. aureus* potentiated IFN-γ–mediated PD-L1 expression in a dose-dependent manner. At 24 h, PD-L1 expression increased by 1.4-fold (*p* = 0.008) and 1.5-fold (*p* < 0.0001) at MOIs of 50 and 200, respectively; corresponding increases at 48 h, were 1.1-fold (*p* = 0.012) and 1.4-fold (*p* < 0.0001) (Appendix A). The potentiating effect was more pronounced in MDA-MB-468 than in MDA-MB-231 and increased with bacterial load.

By contrast, no enhancement of IFN-γ–induced PD-L1 expression was observed in MDA-MB-453 or MCF-12A (Figure 3c, Appendix A), which are less permissive to *S. aureus* internalization.

To assess the durability of this response, we measured PD-L1 expression in MDA-MB-231 cells six days post-infection (Figure 3d, Appendix A). While 24 h stimulation with IFN-γ alone continued to increase PD-L1 expression, the additive effect of *S. aureus* was no longer detectable at this later time point.

Collectively, these results demonstrate that live *S. aureus* can transiently amplify IFN-γ–induced PD-L1 expression in TNBC cell lines with high bacterial uptake. This suggests that intracellular bacteria may act as co-regulators of immune checkpoint signaling by enhancing tumor cell sensitivity to inflammatory cytokines in a context-dependent manner.

### 3.4. S. aureus Enhances IFN-γ–Induced PD-L1 Expression Via STAT1 Activation

To determine whether the enhanced PD-L1 expression observed in infected TNBC cells is mediated Via the JAK/STAT1 pathway, we assessed total STAT1 and phosphorylated STAT1 (p-STAT1) levels in infected and uninfected breast cell lines, with or without IFN-γ stimulation (Figure 4 and Appendix A).

Total STAT1 protein levels remained unchanged across all conditions (Figure 4a,b). As expected, p-STAT1 was undetectable in untreated and *S. aureus*–infected cells in the absence of IFN-γ, indicating that *S. aureus* alone does not activate STAT1. IFN-γ stimulation led to robust p-STAT1 induction in all tested cell lines. Notably, *S. aureus* infection further enhanced IFN-γ–induced STAT1 phosphorylation, with a 2.0-fold (*p* < 0.0001) increase in MDA-MB-468 and a 1.8-fold (*p* < 0.05) increase in MDA-MB-231 compared to IFN-γ treatment alone (Figure 4a,c).

In contrast, no additional p-STAT1 induction was observed in MDA-MB-453 or MCF-12A under co-treatment conditions—cell lines that also showed no enhancement of IFN-γ–mediated PD-L1 expression in response to *S. aureus* infection.

These findings indicate that *S. aureus* infection can potentiate IFN-γ–induced PD-L1 expression Via enhanced STAT1 activation in susceptible TNBC cell lines. This provides mechanistic insight into how intracellular bacteria may modulate immune checkpoint signaling in a cell line–specific manner.

### 3.5. TLR2 Agonists Upregulate PD-L1 Expression in Breast Cell Lines

To determine whether TLR2 signaling influences PD-L1 expression in breast cell lines, we treated the cells with purified *S. aureus* lipoteichoic acid (LTA) and the synthetic TLR2 agonist Pam3CSK4, both with and without IFN-γ stimulation (Figure 5).

We first assessed baseline TLR2 expression by flow cytometry (Figure 5a). MDA-MB-231 cells exhibited markedly higher surface TLR2 expression, followed by MCF-12A, Hs578T, BT-549, MDA-MB-468, and MDA-MB-453. CAL-51 exhibited no detectable TLR2.

We next evaluated whether IFN-γ, a known inducer of PD-L1, also modulates TLR2 levels in selected cell lines (Figure 5b and Appendix A). After 24 h treatment with IFN-γ, a modest but significant increase in TLR2 expression was observed in all cell lines. The most pronounced induction occurred in MDA-MB-231 (1.2-fold; *p* = 0.004), while MDA-MB-468 (1.1-fold; *p* = 0.02), MDA-MB-453 (1.1-fold; *p* = 0.02), and MCF-12A (1.1-fold; *p* = 0.03) showed smaller increases.

We then examined the impact of TLR2 activation on PD-L1 expression. In the absence of IFN-γ, stimulation with 30 µg/mL (but not 1 µg/mL) LTA or 1 µg/mL Pam3CSK4 led to a modest but significant increase in PD-L1 expression in MDA-MB-231 (1.1-fold with LTA and 1.3-fold with Pam3CSK4, respectively; *p* = 0.0097 and *p* < 0.0001), whereas the other cell lines exhibited minimal responses (Figure 5c and Appendix A). These findings suggest that TLR2-mediated PD-L1 induction is more prominent in TLR2-high cells, consistent with prior observations in head and neck squamous cell carcinoma [37].

In the presence of IFN-γ, TLR2 activation further increased PD-L1 levels in all cell lines tested. The greatest enhancement was observed in MDA-MB-453 (1.3-fold increase for both LTA and Pam3CSK4) and MCF-12A (1.3-fold increase for both ligands), followed by MDA-MB-468 (1.1-fold and 1.2-fold) and MDA-MB-231 (1.1-fold and 1.2-fold) (Figure 5c). These results indicate that TLR2 stimulation can augment IFN-γ-driven PD-L1 upregulation in both TNBC and non-malignant breast epithelial cells.

Collectively, these findings demonstrate that bacterial ligands, including *S. aureus*-derived LTA, and synthetic TLR2 agonists can upregulate PD-L1 expression in a TLR2- and cell line–dependent manner. The observed synergy with IFN-γ suggests that TLR2 activation may sensitize breast epithelial cells to inflammatory cytokine signaling, thereby amplifying immune checkpoint expression. These results support a model in which microbial products, in combination with host-derived signals such as IFN-γ, promote immune evasion mechanisms in the tumor microenvironment, particularly in TLR2-expressing TNBCs.

### 3.6. Intracellular S. aureus Clearance Varies in a Cell Line-Dependant Manner

We next assessed intracellular *S. aureus* persistence over a 7-day period post-infection at MOI 50 (Figure 6a) and MOI 200 (Figure 6b) by quantifying viable intracellular bacteria (CFUs) daily. While bacterial loads declined over time in all cell lines, the rate of clearance varied considerably between lines.

At MOI 50, MDA-MB-231 exhibited the slowest clearance (slope: −0.86, 95% CI: −0.92 to −0.80), followed by BT-549 (slope: −1.24, 95% CI: −1.65 to −0.84), MCF-12A (slope: −1.29, 95% CI: −1.41 to −1.18), MDA-MB-453 (slope: −1.38, 95% CI: −1.74 to −1.02), and Hs578T (slope: −1.40, 95% CI: −2.03 to −0.79), with MDA-MB-468 demonstrating the most rapid clearance (slope: −2.07, 95% CI: −2.29 to −1.86) (Figure 6a).

At MOI 200, a similar pattern emerged: MDA-MB-231 again had the slowest clearance (slope: −0.66, 95% CI: −0.73 to −0.59), followed by BT-549 (slope: −0.98, 95% CI: −1.15 to −0.82), MCF-12A (slope: −1.06, 95% CI: −1.18 to −0.94), Hs578T (slope: −1.29, 95% CI: −1.66 to −0.94), and MDA-MB-453 (slope: −1.29, 95% CI: −1.48 to −1.11), with MDA-MB-468 again demonstrating the most efficient clearance (slope: −2.22, 95% CI: −2.74 to −1.70) (Figure 6b). CAL-51 exhibited minimal intracellular CFUs at MOI 200 and none at MOI 50, achieving complete clearance within 24 h (slope: −0.20).

Notably, while *S. aureus* was fully cleared from MDA-MB-231 by day 6 at MOI 50, bacterial persistence extended beyond day 7 at MOI 200. In comparison, all other cell lines cleared viable intracellular bacteria by 3 days at MOI 50 and within 5 days at MOI 200, indicating a cell line–specific and dose-dependent effect on clearance. However, differences in clearance rates between cell lines did not appear to be explained by initial bacterial uptake or burden alone.

These findings suggest potential implications for bacterial survival and host–microbe interactions within the tumor microenvironment. Persistent intracellular *S. aureus* may serve as a reservoir for immunomodulatory stimuli, contributing to immune evasion and influencing therapeutic responses in a cell line–specific manner. The observed variation in clearance kinetics, independent of internalization rates, highlights the complex interplay between host cell-intrinsic immune features and bacterial adaptation in epithelial cancer models.

### 3.7. Intracellular S. aureus Exists in Multiple Forms Within Breast Cell Lines

To further characterize the intracellular fate of *S. aureus*, we performed transmission electron microscopy (TEM) on MDA-MB-231 cells at 24 h and 7 days post-infection (Figure 7).

At 24 h post-infection, *S. aureus* was detected in both the cytoplasm and enclosed within membrane-bound phagosomes (Figure 7a–c). In the cytoplasm, three primary forms were identified: (i) intact coccoid bacteria (Figure 7a), (ii) degraded bacterial remnants (Figure 7a), and (iii) cell wall-deficient L-form bacteria (Figure 7c). Within phagosomes, bacteria exhibited two distinct morphologies: (i) partially degraded or wall-deficient forms (Figure 7a), and (ii) morphologically intact bacteria with visible cell walls (Figure 7a,b).

By 7 days post-infection, intracellular *S. aureus* persisted in four major forms: (i) intact cytoplasmic bacteria (Figure 7d), (ii) L-forms in the cytoplasm (Figure 7d), (iii) viable bacteria within phagosomes (Figure 7e), and (iv) phagosomal bacteria undergoing degradation (Figure 7f). Notably, we observed dividing cocci within phagosomes (Figure 7e), indicating that *S. aureus* retains the capacity for intracellular replication.

These ultrastructural findings corroborate our earlier observations that viable *S. aureus* can be recovered from breast epithelial cells up to 7 days post-infection, and that intracellular persistence can influence host responses, including PD-L1 upregulation. The identification of L-forms and actively replicating bacteria supports the hypothesis that *S. aureus* can evade canonical intracellular clearance mechanisms, possibly contributing to immune evasion and sustained modulation of the tumor microenvironment. This phenotypic plasticity may have important implications for chronic infection, inflammation, and therapeutic resistance in breast cancer.

## 4. Discussion

This study demonstrates that *Staphylococcus aureus* is not merely a bystander in the breast tumor microenvironment but actively invades and persists within triple-negative breast cancer (TNBC) cells in a cell line–dependent manner. We show that intracellular *S. aureus* modulates tumor cell behavior, particularly enhancing IFN-γ–induced PD-L1 expression through TLR2–STAT1 signaling. These effects are contingent on bacterial uptake and vary across TNBC cell lines, underscoring the influence of host cell context in microbial–tumor interactions.

Our gentamicin protection assays and imaging revealed marked variability in *S. aureus* internalization across breast cell lines. MDA-MB-231 and MDA-MB-468 exhibited the highest bacterial uptake, whereas MDA-MB-453, CAL-51, and the non-cancerous MCF-12A line showed low or undetectable internalization. These patterns are not strictly explained by molecular subtype but reflect differences in antimicrobial defenses, bacterial recognition, and receptor expression. This variability highlights the importance of cellular context in shaping bacterial-host interactions, consistent with recent studies emphasizing the influence of cancer cell–intrinsic features on microbial susceptibility [53].

Bacterial uptake likely involves a complex interplay of surface receptors. Although TLR2 is a key sensor of Gram-positive pathogens and can promote bacterial internalization and immune modulation [54,55], we did not observe a direct correlation between baseline TLR2 expression and uptake. For example, MDA-MB-468 cells internalized bacteria efficiently despite low TLR2 levels—suggesting alternative mechanisms such as EGFR-mediated entry may be operative [56,57,58]. Indeed, EGFR, integrins (α5β1, αvβ5), CD36, and microbial ligands such as Eap and Atl may collectively regulate adhesion and invasion [54,55,58,59,60,61,62,63,64].

Persistence of intracellular *S. aureus* was further supported by electron microscopy, which revealed morphologically intact bacteria, degraded forms, and L-forms at both 24 h and 7 days post-infection in MDA-MB-231. L-forms—cell wall–deficient variants previously observed in breast tumors [18]—may contribute to immune evasion and antibiotic resistance [65,66,67,68,69]. Together, these findings point to diverse bacterial survival strategies within tumor cells.

We observed cell line–specific cytotoxic effects following infection, which closely correlated with bacterial internalization. MDA-MB-468 and MDA-MB-231, the most efficient internalizers of *S. aureus*, were also the most sensitive to bacterial challenge, exhibiting acute viability loss and marked reductions in long-term proliferation at high MOIs. Despite this, both lines demonstrated recovery over 7 days, with MDA-MB-231 displaying greater resilience. In contrast, CAL-51, MDA-MB-453, and MCF-12A, which internalized significantly fewer bacteria, were more resistant to cytotoxic effects. These findings highlight intracellular burden as a key determinant of bacterial-induced cytotoxicity and suggest that the therapeutic potential of bacteria-based cancer treatments may depend on tumor-specific features that govern bacterial uptake and intracellular survival. Tumors with higher bacterial internalization capacity may be more susceptible to bacterial-mediated cytotoxicity, while those with low uptake may be less responsive, emphasizing the need for personalized approaches in microbial cancer therapy.

Importantly, purified lipoteichoic acid (LTA) alone had minimal cytotoxic effect, while *S. aureus* exotoxins induced dose-dependent cytotoxicity, particularly in MDA-MB-468 cells. These findings indicate that extracellular toxins—rather than intracellular burden alone—contribute to early cytotoxicity, likely through disruption of calcium homeostasis or membrane integrity [70,71,72,73,74]. Nonetheless, cell survival and recovery highlight the dynamic nature of host–microbe interactions in TNBC.

A key novel finding is that intracellular *S. aureus* enhances IFN-γ–induced PD-L1 expression in a cell line–specific and dose-dependent manner. This effect was observed in MDA-MB-468 and MDA-MB-231, both of which support high bacterial uptake. In contrast, low-internalizing lines such as MDA-MB-453 and MCF-12A showed no PD-L1 modulation. These results suggest that intracellular bacterial burden is required to potentiate IFN-γ–mediated immune signaling.

Mechanistically, *S. aureus* co-treatment increased STAT1 phosphorylation in the presence of IFN-γ, supporting a model in which bacterial sensing enhances canonical interferon responses. TLR2 agonists (Pam3CSK4 and LTA) modestly increased PD-L1 only in the high-TLR2 line, MDA-MB-231, and this effect was amplified by IFN-γ, consistent with prior findings in head and neck cancer [36,37]. Notably, IFN-γ stimulation upregulated TLR2 in all cell lines, potentially priming them for bacterial ligand responsiveness.

These data support a two-step mechanism: IFN-γ increases TLR2 expression, and subsequent exposure to bacterial ligands amplifies PD-L1 expression through TLR2–STAT1 signaling. However, we did not directly assess the roles of NF-κB, STING, or other pattern recognition receptors. Prior reports implicate STING in PD-L1 upregulation Via type I IFNs [75,76,77]; given that *S. aureus* DNA may activate STING, further mechanistic studies are warranted.

Our findings contribute to a growing appreciation of the dual roles of intratumoral bacteria. On one hand, microbial modulation of immune checkpoints could impair anti-tumor immunity by enhancing adaptive resistance. On the other hand, bacteria—especially when engineered—may be leveraged therapeutically to deliver immune-stimulating agents, induce tumor lysis, or reshape the tumor microenvironment in favor of immunotherapy [28,29,30,31].

Previous studies have shown that *S. aureus*–derived α-hemolysin and enterotoxins can trigger apoptosis in tumor cells and promote immune activation [38,72,73]. Conversely, virulence factors such as FnBPA and LTA may support tumor growth Via TLR signaling and cytokine induction [70,71]. Our results suggest that *S. aureus* has the capacity to modulate both tumor viability and immune signaling in TNBC, depending on bacterial load and host cell properties.

These findings also raise questions about how the tumor microbiome might influence response to immune checkpoint inhibitors (ICIs). While *S. aureus*–induced PD-L1 upregulation could enhance ICI efficacy in PD-L1–low tumors, it might also contribute to resistance by reinforcing immune evasion. A deeper understanding of microbial–immune–tumor crosstalk is needed to clarify these effects.

Several limitations should be acknowledged. First, while we observed that *S. aureus* enhanced IFN-γ–induced PD-L1 expression in a subset of TNBC cell lines, we did not directly test the causal involvement of the TLR2–STAT1 pathway using knockdown, blocking antibodies, or pharmacologic inhibitors. Thus, our mechanistic conclusions remain inferential and require confirmation in targeted perturbation studies. Second, although we included a broad panel of breast cell lines representing the major TNBC subtypes and a non-tumorigenic control, we did not assess IFN-γ–induced PD-L1 modulation in Hs578T or additional mesenchymal lines. A comprehensive analysis would require a prespecified panel of subtype-matched models tested in parallel, which is beyond the scope of this manuscript but will be pursued in future work. Third, we used high MOIs to achieve consistent intracellular infection, which may not reflect physiological exposure levels. Finally, our experiments were conducted in vitro; validation in primary cultures, co-culture systems, and in vivo models will be critical to establish the translational relevance of these findings.

From a translational perspective, these findings suggest that bacterial profiling of tumors could inform immunotherapy response. Differences in microbial colonization, TLR2 expression, and bacterial responsiveness among TNBC subtypes may create unique vulnerabilities or resistance mechanisms. Strategies to target bacterial–TLR2–PD-L1 signaling—whether Via antibiotics, microbiome modulation, or engineered bacteria—may enhance ICI efficacy in selected patients.

While antibiotics remain the frontline approach for infection control, their impact on tumor-associated microbiota is complex. Broad-spectrum antibiotics may impair ICI efficacy by disrupting beneficial microbial–immune interactions [78,79,80,81]. Conversely, probiotics and fecal microbiota transplantation have shown promise in enhancing immunotherapy responsiveness, especially with microbiota enriched in *Akkermansia muciniphila* and *Bifidobacterium* species [82]. Whether similar approaches apply to breast cancer remains to be determined.

## 5. Conclusions

In summary, we show that *Staphylococcus aureus* invades and persists in triple-negative breast cancer cells, modulates IFN-γ–induced PD-L1 Via TLR2–STAT1 signaling, and alters tumor cell viability in a cell line–specific manner. These findings offer novel insight into host–microbe–tumor interactions and provide a foundation for future exploration of bacterial modulation as a therapeutic target or adjunct to immunotherapy in TNBC.

## Figures and Tables

**Figure 1 cancers-17-02947-f001:**
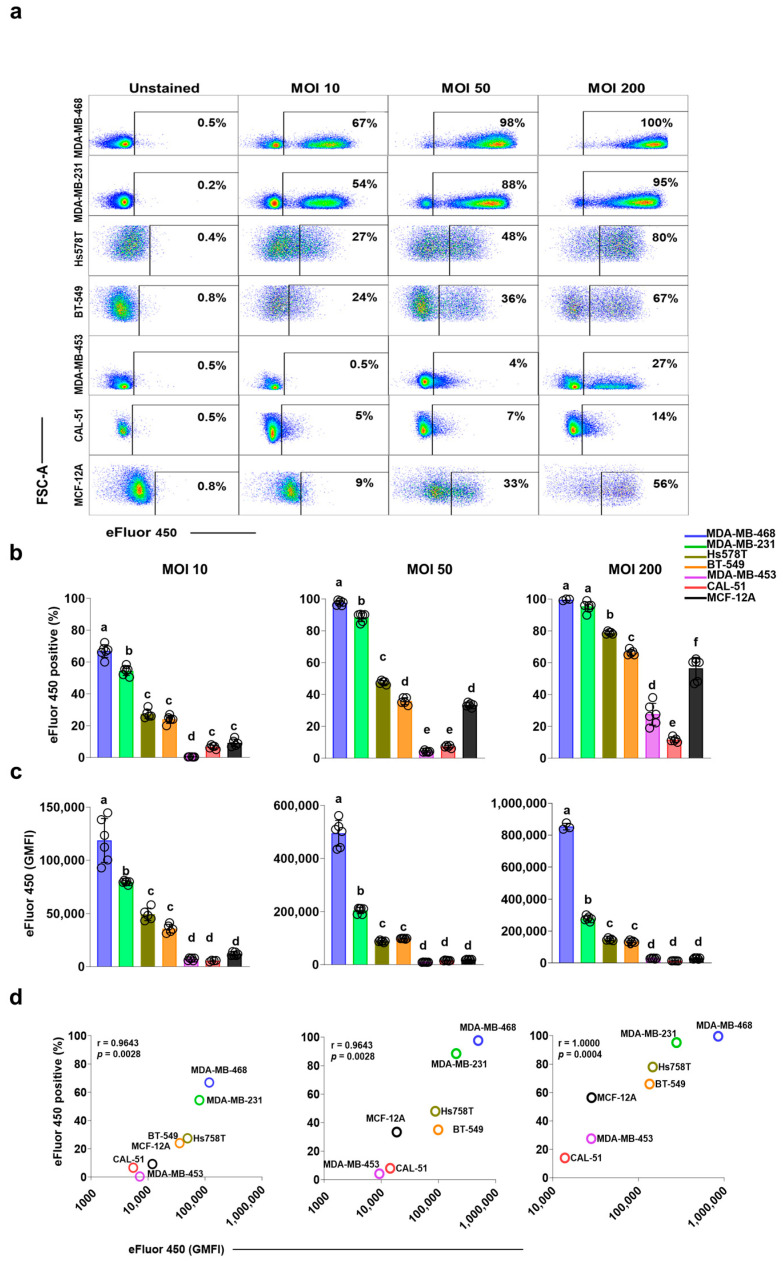
Internalization of *S. aureus* in triple-negative breast cancer and non-cancerous breast cell lines (**a**) Representative flow cytometry scatter plots depicting the percentage of viable eFluor450-positive cells, indicating internalized *S. aureus* in the breast cell lines at 24 h post-infection. Cells were incubated with eFluor450-labeled *S. aureus* for 2 h at multiplicities of infection (MOIs) of 10, 50, and 200, followed by washing and gentamicin treatment to eliminate extracellular bacteria. (**b**) Quantification of the percentage of infected cells across the different cell lines, highlighting differences in *S. aureus* internalization efficiency. (**c**) Geometric mean fluorescence intensity (GMFI) of eFluor450 in infected cells, representing the intracellular bacterial burden per cell. Statistical comparisons in panels (**b**,**c**) were performed using a mixed-effects model with Tukey’s multiple comparisons test. Data are presented as individual values ± standard deviations (SD) from at least three independent experiments. Different letters denote statistically significant differences between cell lines (*p* < 0.0001). Full statistical details are provided in Appendix A. (**d**) Spearman’s correlation between the percentage of infected cells and intracellular bacterial burden at MOIs of 10, 50, and 200, demonstrating a dose-dependent relationship between *S. aureus* internalization and bacterial load per cell.

**Figure 2 cancers-17-02947-f002:**
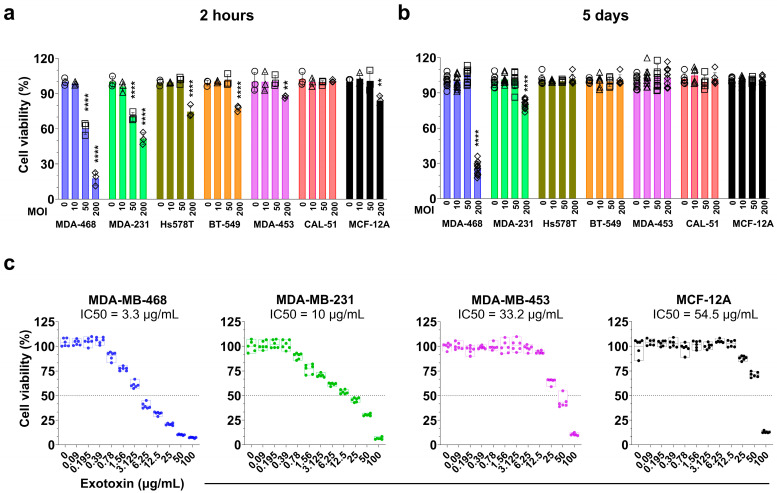
Cytotoxicity and inhibition of proliferation induced by *S. aureus* infection and exotoxins in breast cell lines. (**a**) Viability of breast cells 2 h post-infection with viable *S. aureus* at MOIs 10, 50, and 200. (**b**) Long-term effects of *S. aureus* infection on cell proliferation measured using crystal violet assay, normalizing to uninfected cells. One-way ANOVA with Tukey’s multiple comparisons test was performed to compare between infected cells with different MOIs (10, 50, and 200) and uninfected cells (MOI 0) within each cell line. Data represents the mean ± SD from at least three independent experiments. Statistical significance: ** *p* < 0.01; **** *p* < 0.0001. (**c**) Dose-dependent cytotoxicity of total *S. aureus* exotoxins in breast cells after 5 days using crystal violet assay normalizing to uninfected cells. IC50 values for each cell line are indicated in the figure.

**Figure 3 cancers-17-02947-f003:**
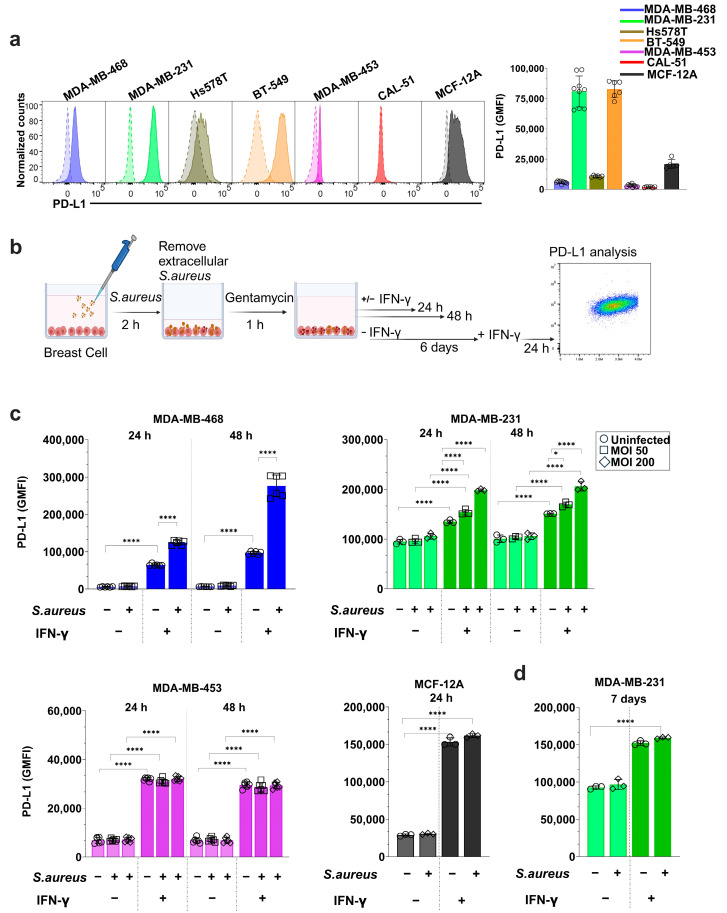
PD-L1 surface expression and the impact of intracellular *S. aureus* in breast cell lines (**a**) Baseline PD-L1 expression in TNBC cell lines (MDA-MB-468, MDA-MB-231, Hs578t, BT-549, MDA-MB-453 and CAL-51) and the non-cancerous epithelial cell line (MCF-12A), presented as representative histograms (**left**) and calculated geometric mean fluorescence intensity (GMFI) values (**right**). Unstained controls are shown as dashed lines, and PD-L1-stained cells as solid lines. (**b**) Schematic overview of the experimental design for bacterial infection and cytokine stimulation. (**c**) PD-L1 surface expression following infection with *S. aureus* at 24 and 48 h, with or without IFN-γ stimulation (5 µM). Cell lines were infected at different multiplicities of infection (MOI): MDA-MB-468 at MOI 50; MDA-MB-231 and MDA-MB-453 at MOI 50 or 200; MCF-12A at MOI 200, based on infection efficiency and viability. (**d**) PD-L1 expression in MDA-MB-231 cells 7 days post-infection. Cells were infected with *S. aureus*, cultured without IFN-γ for 6 days, and then stimulated with IFN-γ (5 µM) for 24 h prior to analysis. One-way ANOVA with Tukey’s multiple comparisons was used for statistical analysis. Statistics for relevant comparison were presented in the figures, and full statistics for all comparisons can be found in Appendix A. Data represent the mean ± SD from at least three independent experiments. Statistical significance: * *p* < 0.05, *****p* < 0.0001.

**Figure 4 cancers-17-02947-f004:**
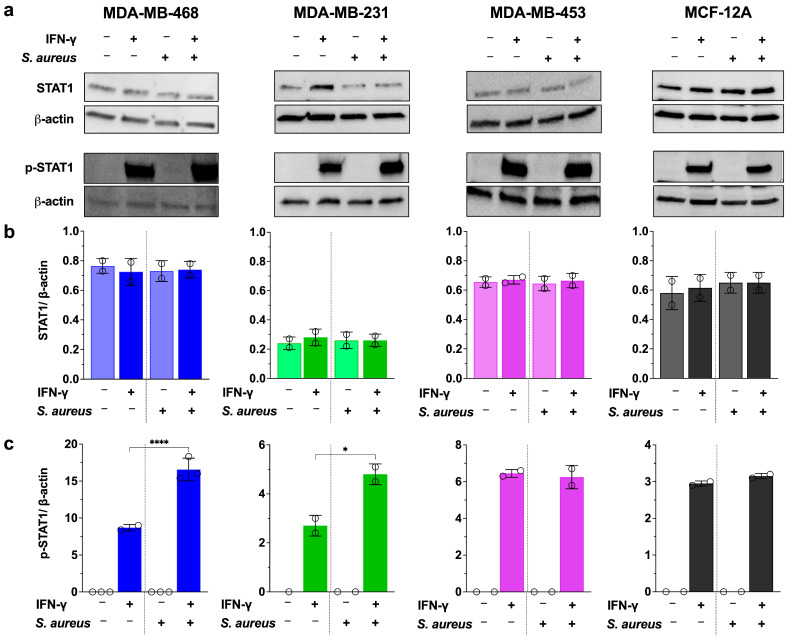
*S. aureus* infection enhances IFN-γ-induced STAT1 phosphorylation in breast cell lines. (**a**) Western blot analysis of total STAT1 and phosphorylated STAT1 (p-STAT1(Y701)) expression in breast cell lines treated with or without IFN-γ, following *S. aureus* infection at MOI 50 for MDA-MB-468 and MDA-MB-231 and at MOI 200 for MDA-MB-453 and MCF-12A. Cells were infected with *S. aureus* for 2 h and then treated with 5 µM IFN-γ for 24 h. Total protein was extracted and analyzed for p-STAT1, STAT1, and β-actin as a loading control. (**b**) Total STAT1 levels remain unchanged across all conditions. (**c**) Quantification of relative p-STAT1 levels shows a significant increase with IFN-γ treatment, which is further enhanced in *S. aureus*-infected MDA-MB-468 and MDA-MB-231 treated with IFN-γ. Data represent mean ± SD from two to three independent experiments. Statistical significance: * *p* > 0.05; **** *p* < 0.0001. The full uncropped blots are shown in Appendix A.

**Figure 5 cancers-17-02947-f005:**
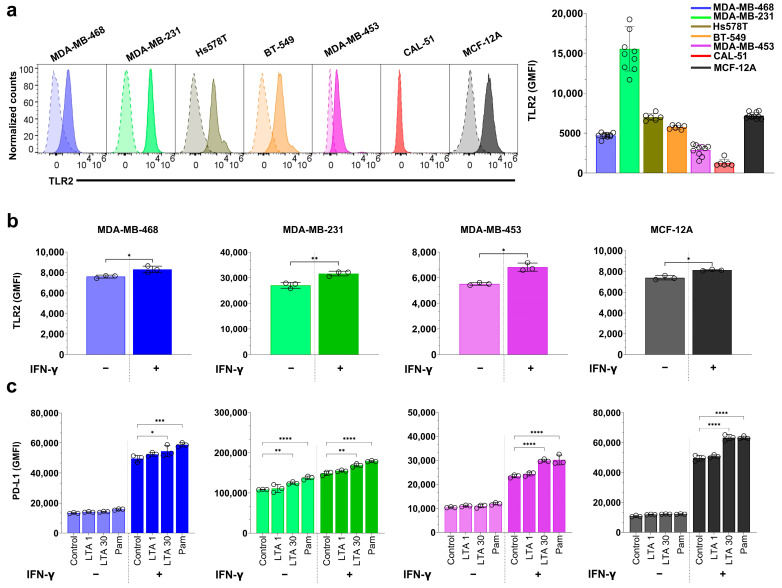
Baseline TLR2 expression and TLR2 agonist-induced PD-L1 upregulation in breast cell lines. (**a**) Representative flow cytometry histograms of baseline TLR2 expression in breast cell lines. Unstained controls are shown as dashed lines, and TLR2-stained cells as solid lines, with corresponding GMFI values displayed to indicate relative expression levels (**b**) TLR2 GMFI in cell lines following 24 h treatment with 5 µM IFN-γ. (**c**) PD-L1 GMFI in cell lines following 24 h treatment with 1 µg/mL or 30 µg/mL purified *S. aureus* LTA or 1 µg/mL Pam3CSK4, with or without 5 µM IFN-γ-stimulation. Statistical analysis was performed using one-way ANOVA with Tukey’s multiple comparisons test. Data are presented as mean ± SD from at least three independent experiments. Statistical significance: * *p* < 0.05; ** *p* < 0.01; *** *p* < 0.001; **** *p* < 0.0001.

**Figure 6 cancers-17-02947-f006:**
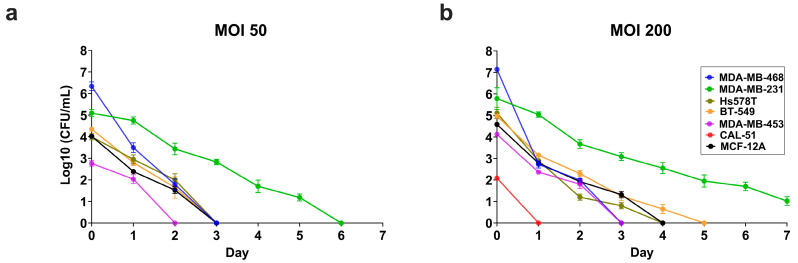
Intracellular viable *S. aureus* clearance in triple-negative breast cancer and non-cancerous breast epithelial cell lines. Persistence of viable intracellular *S. aureus* at (**a**) MOI 50 and (**b**) MOI 200 was assessed by quantifying colony-forming units (CFUs) from cell lysates over 7 days. A pseudo-count of 1 was added to indicate the absence of bacterial growth. Bacterial clearance rates were estimated using simple linear regression, and two-tailed *t*-tests were performed to compare slopes between cell lines. All data are presented as mean ± SD of three replicates.

**Figure 7 cancers-17-02947-f007:**
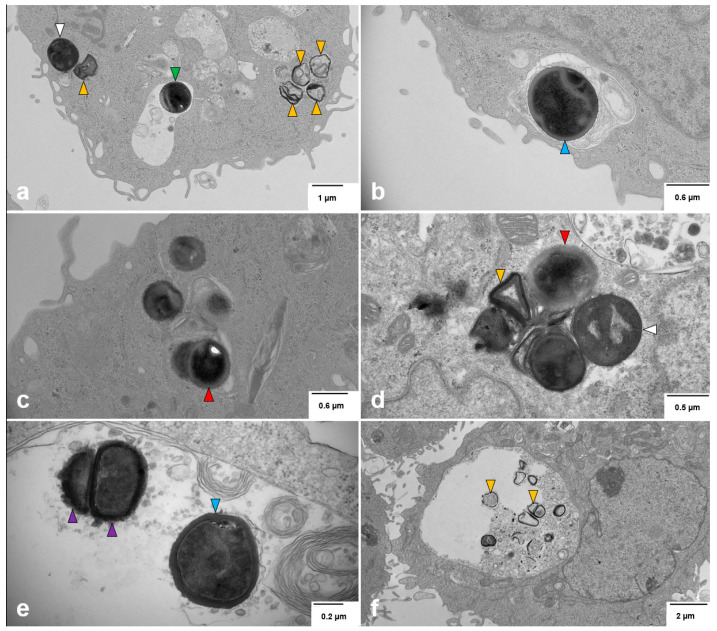
Morphology of intracellular *S. aureus* in TNBC Cells. Transmission electron microscopy images of intracellular *S. aureus* in MDA-MB-231 cells at 24 h (**a**–**c**) and 7 days (**d**–**f**) post-infection. (**a**) *S. aureus* within the cytoplasm (white arrowhead), appearing as dense, spherical structures with a well-defined cell wall, characteristic of Gram-positive cocci. Some intact bacteria are enclosed within a phagosome (green arrowhead), where a visible separation between the inner cytoplasmic membrane and the thick peptidoglycan layer suggests bacterial adaptation to the intracellular environment. The surrounding host cytoplasm contains intracellular vacuoles and membrane compartments, indicative of potential bacterial containment within a phagosomal or vacuolar compartment. Degraded bacterial remnants dispersed in the cytoplasm (orange arrowheads) suggest partial bacterial degradation. (**b**) *S. aureus* enclosed within a phagosome (blue arrowhead), maintaining its characteristic coccoid shape and thick peptidoglycan layer, indicative of intracellular persistence. (**c**) L-form *S. aureus* (red arrowhead) within the cytoplasm, distinguished by an irregular shape and the absence of a rigid cell wall. This morphotype suggests bacterial adaptation to intracellular stress, potentially facilitating immune evasion and antibiotic resistance. (**d**) A mixture of bacterial morphotypes 7 days post-infection, including intact coccoid bacteria (white arrowhead), L-forms (red arrowhead), and partially degraded bacteria (orange arrowhead), highlighting the diverse survival strategies of intracellular *S. aureus*. (**e**) Intact *S. aureus* (blue arrowhead) and actively dividing bacteria within a phagosome (purple arrowheads), indicating ongoing bacterial replication despite the intracellular environment. (**f**) Degraded bacterial remnants within a phagosome (orange arrowheads), suggesting host-mediated bacterial breakdown, though complete clearance appears incomplete. These images illustrate the morphological diversity of *S. aureus* within TNBC cells, emphasizing bacterial adaptation, survival, and persistence over time.

## Data Availability

The raw data supporting the conclusions of this article will be made available by the authors on request.

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
