# Peer review of "Cell Line-Dependent Internalization, Persistence, and Immunomodulatory Effects of Staphylococcus aureus in Triple-Negative Breast Cancer"

_cancers, 2025, doi:10.3390/cancers17182947_

Round 1

Reviewer 1 Report (Previous Reviewer 2)

Comments and Suggestions for Authors

The submitted article entitled “Cell Line-Dependent Internalization, Persistence, and Immunomodulatory Effects of Staphylococcus aureus in Triple-Negative Breast Cancer” by Kianpour Rad et al. demonstrated the role of S. aureus in influencing the immune milieu of TNBC cells that have a potential for improving the therapeutic efficacy. To establish the above hypothesis, authors have used series of TNBC cell lines, where they have shown intricately the differential capacity of S.aureus internalization and effect of the inoculation on cell viability. The authors have also reported the increase of IFN ɣ- induced PDL1 in S. aureus infected cells compared to the basal level expression and have tried to elucidate the mechanism of PDL1 increase related to phosphorylation of STAT. Finally, the authors tried to study the expression of IFN ɣ-induced PDL1 in relation to TLR2 with Lipoteichoic acid in TNBC cells.

General Critique:

The study is novel, impressive, and can be considered for publication. The authors have conducted additional experiments that are satisfactory and explain the present title. However, some experimental designs are required to be added for publication along with minor corrections.

  1. In Figure 1 the authors demonstrate the efficiency of internalization of S. aureus in different TNBC cell lines. MDA-MB-468 and MDA-MB-231 showed high efficiency followed by Hs578T and BT-549, the lowest being MDA-MB-453 and CAL-51. Interestingly, the authors show in Figure 3C, PDL1expression increases significantly in MDA-MB-468 from basal level, MDA-MB231 being modest (as surface level PDL1- is high itself). However, it’s very interesting to investigate the effect of Hs-578T, where PDL1 basal level is low again (similar to MDA-MB-468) and the internalization capacity is also moderate (48%). The experimental result can be included in the additional figure file if required.

Minor Critique:

  1. Figure 3C, Figure 5B and C- Should be represented on the same scale and by line graph for better understanding the comparison between the cell lines.
  2. Same scale representation should be done in Figure 4B and C, to understand the comparison clearly.

Author Response

Comment: The submitted article entitled “Cell Line-Dependent Internalization, Persistence, and Immunomodulatory Effects of Staphylococcus aureus in Triple-Negative Breast Cancer” by Kianpour Rad et al. demonstrated the role of S. aureus in influencing the immune milieu of TNBC cells that have a potential for improving the therapeutic efficacy. To establish the above hypothesis, authors have used series of TNBC cell lines, where they have shown intricately the differential capacity of S.aureus internalization and effect of the inoculation on cell viability. The authors have also reported the increase of IFN ɣ- induced PDL1 in S. aureus infected cells compared to the basal level expression and have tried to elucidate the mechanism of PDL1 increase related to phosphorylation of STAT. Finally, the authors tried to study the expression of IFN ɣ-induced PDL1 in relation to TLR2 with Lipoteichoic acid in TNBC cells.

General Critique:

The study is novel, impressive, and can be considered for publication. The authors have conducted additional experiments that are satisfactory and explain the present title. However, some experimental designs are required to be added for publication along with minor corrections.

Response: We thank Reviewer 1 for the careful reading of our revised manuscript and for the constructive comments. We have addressed each point in detail below, with corresponding revisions in the manuscript.

Comment 1: In Figure 1 the authors demonstrate the efficiency of internalization of S. aureus in different TNBC cell lines. MDA-MB-468 and MDA-MB-231 showed high efficiency followed by Hs578T and BT-549, the lowest being MDA-MB-453 and CAL-51. Interestingly, the authors show in Figure 3C, PDL1expression increases significantly in MDA-MB-468 from basal level, MDA-MB231 being modest (as surface level PDL1- is high itself). However, it’s very interesting to investigate the effect of Hs-578T, where PDL1 basal level is low again (similar to MDA-MB-468) and the internalization capacity is also moderate (48%). The experimental result can be included in the additional figure file if required.

Response 1: We thank the reviewer for the thoughtful suggestion to include PD-L1 data for Hs578T given its low basal PD-L1 and moderate S. aureus internalization. A rigorous assessment would require a prespecified hypothesis and an expanded TNBC panel (additional mesenchymal lines and matched controls), with infection ± IFN-γ performed in parallel batches and accompanied by targeted pathway perturbations (e.g., TLR2/STAT1). This constitutes a separate, focused study and is outside the scope of the present manuscript, which is already extensive. To avoid overinterpretation, we have narrowed the claims to the lines tested and added a limitations note to the Discussion indicating that this question will be pursued in a follow-up study: “while we observed that S. aureus enhanced IFN-γ–induced PD-L1 expression in a subset of TNBC cell lines, we did not directly test the causal involvement of the TLR2–STAT1 pathway using knockdown, blocking antibodies, or pharmacologic inhibitors. Thus, our mechanistic conclusions remain inferential and require confirmation in targeted perturbation studies. Second, although we included a broad panel of breast cell lines representing the major TNBC subtypes and a non-tumorigenic control, we did not assess IFN-γ–induced PD-L1 modulation in Hs578T or additional mesenchymal lines. A comprehensive analysis would require a prespecified panel of subtype-matched models tested in parallel, which is beyond the scope of this manuscript but will be pursued in future work. Third, we used high MOIs to achieve consistent intracellular infection, which may not reflect physiological exposure levels. Finally, our experiments were conducted in vitro; validation in primary cultures, co-culture systems, and in vivo models will be critical to establish the translational relevance of these findings.”

Minor Critique:

Comment 2: Figure 3C, Figure 5B and C- Should be represented on the same scale and by line graph for better understanding the comparison between the cell lines.

Response 2: We thank the reviewer for this suggestion. We note that baseline expression data are already presented on standardized axes (Figure 3A for PD-L1, Figure 5A for TLR2), allowing direct comparison of absolute levels across all cell lines. Figures 3C–D and 5B–C, by contrast, are intended to illustrate treatment-dependent changes within each cell line. Because expression levels vary substantially between models (e.g., TLR2 GMFI in MDA-MB-231 is ~15,000–30,000 versus ~5,000–8,000 in MDA-MB-468; PD-L1 in MDA-MB-468 exceeds 200,000 GMFI while MCF-12A remains <20,000), forcing a common axis would compress lower-expressing lines and obscure meaningful induction, while exaggerating variability in higher-expressing lines. For this reason, we retained optimized scales for each panel but clearly report fold-changes and statistical comparisons to facilitate interpretation.

We also note that a line graph format is not appropriate for these data, as the conditions represent categorical treatments rather than continuous variables or paired data. The current grouped bar graphs with individual data points are the most accurate and transparent way to convey treatment effects and biological differences without misrepresentation.

Comment 3: Same scale representation should be done in Figure 4B and C, to understand the comparison clearly.

Response 3: We thank the reviewer for this suggestion. We agree that Figure 4B (STAT1/β-actin) can be presented on a uniform y-axis, as values across cell lines fall within a similar range (~0.2–1.0). We have revised this panel accordingly. For Figure 4C (pSTAT1/β-actin), however, expression varies much more widely (0–20), and forcing a common axis would obscure meaningful induction in lower-responding lines while compressing the dynamic range in higher-responding lines. To preserve interpretability, we have retained optimized axes in Figure 4C while ensuring fold-changes and statistical comparisons are clearly indicated.

Reviewer 2 Report (New Reviewer)

Comments and Suggestions for Authors

This is a revised manuscript. It has been modified a lot; however, some points have to be fixed.

  • I strongly recommend that the authors add numerical results in the abstract. Moreover, a graphical abstract is required in my view since this manuscript is very sophisticated.
  • In the last paragraph of the introduction ( lines 121-132), the authors described a brief of their methodology, results, and comparison with their previous findings. This looks like a conclusion, but the authors should describe their aim of this work. The authors may move it to the conclusion and write a paragraph about the aim of this work.
  • Why did the authors use some anonymous cell lines? The authors should write the reason in the Materials and Methods. ( CAL-51, BT-549, HS578T)
  • The authors should write the reason for labeling the bacteria with eFluor450.
  • How the crystal violet staining assay measured the effect of aureus infection on cell proliferation ( lines 217-221). It is generally assumed that crystal violet staining is used to distinguish between Gram-negative and Gram-positive bacteria.
  • In Figs 2a and b, it seems that there is no difference between the cell viability of the cell lines infected by aureus in two hours and five days. How do the authors justify this phenomenon?
  •  In section 3.3 ( line 432), the authors claimed aureus infection enhances INF-ꝩ. I wonder if this experiment has been done in animal models, but it was not mentioned.
  • Is aureus intracellular (section 3.7), as long as in microbiology and clinical texts it has been considered an extracellular pathogen.

Author Response

Comment: This is a revised manuscript. It has been modified a lot; however, some points have to be fixed.

Response: We thank Reviewer 2 for the careful reading of our revised manuscript and for the constructive comments. We have addressed each point in detail below, with corresponding revisions in the manuscript.

Comment 1: I strongly recommend that the authors add numerical results in the abstract. Moreover, a graphical abstract is required in my view since this manuscript is very sophisticated.

Response1: We thank the Reviewer for this helpful suggestion. The Abstract has been revised to incorporate key numerical results, including:

Internalization rates (e.g., S. aureus internalized into 67% of MDA-MB-468 and 54% of MDA-MB-231 cells at MOI 10, compared with only 0.5–9% in low-uptake lines).

Cytotoxicity and proliferation data (e.g., 82% reduction in MDA-MB-468 viability at 2 h and 74% decrease in proliferation after 5 days).

Persistence of intracellular bacteria (up to 7 days in MDA-MB-231, versus clearance within 3–5 days in other cell lines).

PD-L1 induction (e.g., co-treatment with IFN-γ and S. aureus increased PD-L1 up to 2.9-fold in MDA-MB-468 and 1.5-fold in MDA-MB-231).

We also now specify in the Abstract that bacteria were labeled with eFluor450 for flow cytometry.

We have made additional edits to reduce the abstract word count to around 250, as required by the journal.

Comment 2: In the last paragraph of the introduction (lines 121-132), the authors described a brief of their methodology, results, and comparison with their previous findings. This looks like a conclusion, but the authors should describe their aim of this work. The authors may move it to the conclusion and write a paragraph about the aim of this work.

Response 2: We thank the Reviewer for this helpful observation. We have revised the end of the Introduction to remove text that read like a conclusion or preview of the results. Instead, we now clearly state the aims of this work. Specifically, the Introduction now highlights the knowledge gap regarding whether viable S. aureus can invade and persist in TNBC cells and modulate PD-L1 expression. We then introduce the rationale for our chosen panel of cell lines, followed by a concise aims paragraph that outlines our study objectives: to determine the ability of S. aureus to invade and persist in TNBC cells, assess its cytotoxic and proliferative effects, and evaluate whether intracellular bacteria modulate PD-L1 via TLR2/STAT1 signaling in the context of IFN-γ stimulation. This restructuring addresses the Reviewer’s concern and improves the logical flow of the Introduction.

Comment 3: Why did the authors use some anonymous cell lines? The authors should write the reason in the Materials and Methods. (CAL-51, BT-549, HS578T)

Response 3: We thank the Reviewer for pointing this out. By “anonymous,” we understand the Reviewer to mean less frequently used TNBC cell lines such as CAL-51, BT-549, and Hs578T. We included these models because they represent distinct TNBC molecular subtypes (mesenchymal or mesenchymal stem-like), as defined by Lehmann et al. (JCI, 2011). Together with MDA-MB-231, MDA-MB-468, and MDA-MB-453, this panel captures the major transcriptional subtypes of TNBC. These lines also differ in baseline PD-L1 expression, epithelial–mesenchymal phenotype, and innate immune signaling capacity, providing a more comprehensive system to evaluate host–microbe interactions.

We also note that inclusion of additional TNBC lines was suggested by the reviewers during the initial round of peer review, and we have incorporated this feedback to strengthen the study.

We have now clarified this rationale in the appropriate sections (2.2) of the Materials and Methods: “These cell lines were selected to represent the major molecular subtypes of TNBC (basal-like, mesenchymal/mesenchymal stem-like, and luminal androgen receptor), which differ in PD-L1 expression, epithelial–mesenchymal phenotype, and innate immune signaling capacity. This diversity provided a comprehensive model system to investigate host–microbe interactions.”

This is now in addition to what was stated in the Induction “To address this, we employed a panel of well-characterized breast cell lines repre-sentative of key TNBC subtypes: MDA-MB-468 (basal-like 1), MDA-MB-231 and Hs578T (mesenchymal stem-like), BT-549 and CAL-51 (mesenchymal), and MDA-MB-453 (lu-minal androgen receptor), alongside the non-tumorigenic mammary epithelial line MCF-12A. These models differ in baseline PD-L1 expression, epithelial–mesenchymal phenotype, and innate immune signaling capacity, providing a valuable system to dissect host–microbe interactions.”

Comments 4: The authors should write the reason for labeling the bacteria with eFluor450.

Response 4: We thank the reviewer for this helpful comment. S. aureus was labeled with eFluor450 to enable sensitive detection of bacterial internalization by flow cytometry. eFluor450 was chosen for its bright, stable signal in the violet channel, which minimizes spectral overlap with other fluorophores and reduces host cell autofluorescence. This ensured accurate quantification of bacterial uptake across the breast cell line panel.

We have revised section 2.3 of the Methods to clarify this rationale: “eFluor450 was selected for its bright and stable signal in the violet channel, which enables sensitive detection of bacteria by flow cytometry while minimizing spectral overlap and host cell autofluorescence.”

Comment 5: How the crystal violet staining assay measured the effect of aureus infection on cell proliferation ( lines 217-221). It is generally assumed that crystal violet staining is used to distinguish between Gram-negative and Gram-positive bacteria.

Response 5: We thank the reviewer for this comment and the opportunity to clarify. In this study, the crystal violet assay was not used for Gram differentiation, but as a colorimetric method to assess host cell proliferation following S. aureus infection. Crystal violet binds to DNA and proteins of adherent cells; after solubilization, the retained dye is quantified spectrophotometrically as a surrogate measure of host cell density and growth. Although crystal violet can also stain bacterial biomass, in our assay this contribution is negligible because extracellular bacteria were removed by gentamicin treatment and the remaining internalized bacteria represent only a minute fraction compared with the host cell biomass. Thus, the absorbance signal primarily reflects the number of adherent mammalian cells, providing a reliable readout of cell viability and proliferation. We have revised the Methods section (2.6) to make this purpose explicit: “Crystal violet staining was used to quantify adherent host cell biomass as a proxy for viability and proliferation. Although the dye can also stain bacteria, the contribution of internalized organisms is negligible compared with host cell content due to prior gentamicin treatment and the low bacterial load. Thus, the absorbance measurement primarily reflects mammalian cell proliferation.”

Comment 6: In Figs 2a and b, it seems that there is no difference between the cell viability of the cell lines infected by aureus in two hours and five days. How do the authors justify this phenomenon?

Response 6: We thank the reviewer for this observation. The similarity between the 2-hour (Fig. 2a) and 5-day (Fig. 2b) results arises because the same factors that drive acute cytotoxicity also determine longer-term proliferative capacity. Lines that were resistant to early S. aureus–induced cytotoxicity (e.g., MDA-MB-453, CAL-51, MCF-12A) expanded normally when reseeded, whereas those that were acutely sensitive (MDA-MB-468, and to a lesser extent MDA-MB-231 at MOI 200) showed impaired proliferation over 5 days. Thus, acute cytotoxicity and 5-day growth inhibition mirror each other, with long-term outcomes largely dictated by the extent of early cell loss and stress burden. This concordance is consistent with prior observations that acute injury often predicts long-term proliferative impairment, both in bacterial infection models (PMID: 22919634) and in standard clonogenic survival assays of radiation and cytotoxic drugs (PMID: 17406473).

Comment 7: In section 3.3 ( line 432), the authors claimed aureus infection enhances INF-ꝩ. I wonder if this experiment has been done in animal models, but it was not mentioned.

Response 7: We thank the reviewer for the question. To clarify, our study did not measure IFN-γ, and we did not claim that S. aureus infection increased IFN-γ in our system. In the Discussion we wrote: “While IFN-γ is a well-characterized inducer of PD-L1, previous studies have shown that bacterial components such as heat-killed S. aureus can stimulate IFN-γ-producing lymphocytes and enhance PD-L1 expression in cancer models.” This sentence was intended as context, not as a report of new in vivo data from our work.

Consistent with the reviewer’s request for animal evidence, prior in vivo studies show that S. aureus infection elicits IFN-γ responses in mice (PMIDs: 21603642, 23792139, 10768926, 18606687). Separately, bacterial products can increase tumor PD-L1 in vivo; for example, lipopolysaccharide (LPS) upregulates PD-L1 in murine pancreatic ductal adenocarcinoma and hepatocellular carcinoma models (PMIDs: 34718325, 35438468). We have clarified this point in section 3.3 to avoid any implication that we measured IFN-γ in our experiments.

Comment 8: Is aureus intracellular (section 3.7), as long as in microbiology and clinical texts it has been considered an extracellular pathogen.

Response 8: We appreciate the reviewer’s point. S. aureus is classically described as an extracellular pathogen in many clinical texts; however, a large body of work shows that it can invade and persist inside professional and non-professional phagocytes—hence it is now often termed a facultative intracellular pathogen. Mechanistically, staphylococcal fibronectin-binding proteins (FnBPs) bridge host fibronectin to α5β1-integrins on epithelial/endothelial cells to trigger uptake, and intracellular persistence (including small-colony-variant phenotypes) has been documented in vitro and in vivo (PMIDs: 11207545, 10456915, 9826383, 22919634, 8627043). These intracellular niches are increasingly recognized as relevant to chronic/recurrent infection and antibiotic tolerance (PMID: 22919634).

In our study, we use the term “intracellular” in this well-established sense—i.e., bacteria that have been internalized by breast epithelial/cancer cells. Experimentally, we quantified internalization using eFluor450-labeled S. aureus and post-infection antibiotic steps to eliminate extracellular bacteria prior to downstream readouts, consistent with standard approaches in the field.

Round 2

Reviewer 1 Report (Previous Reviewer 2)

Comments and Suggestions for Authors

The submitted article entitled “Cell Line-Dependent Internalization, Persistence, and Immunomodulatory Effects of Staphylococcus aureus in Triple-Negative Breast Cancer” by Kianpour Rad et al. reports the role of S. aureus in influencing the immune milieu of TNBC cells that have a potential for improving the therapeutic efficacy. To establish the above hypothesis, authors have used a batch of TNBC cell lines where they have shown the increase of IFN ɣ- induced PDL1 in S. aureus infected cells. They also tried to elucidate the mechanism of PDL1 increase related to phosphorylation of STAT. Finally, the authors tried to study the expression of IFN ɣ-induced PDL1 in relation to TLR2 with Lipoteichoic acid in TNBC cells.

The study is well equipped with novel and strong theoretical base. The author’s responses to the reviewer’s concerns were also satisfactorily supported. Hence, I strongly recommend accepting this article in its current form for this journal.

Reviewer 2 Report (New Reviewer)

Comments and Suggestions for Authors

The authors answered my comments well, and I recommend publishing. 

This manuscript is a resubmission of an earlier submission. The following is a list of the peer review reports and author responses from that submission.

Round 1

Reviewer 1 Report

Comments and Suggestions for Authors

The authors are silent on another tumor -targeting bacteria that also affects the immune system (PMID: 15644448; PMID: 16885365; PMID: 17548809; PMID: 19199339; PMID: 19221501; PMID: 19528442; PMID: 19766244; PMID: 21135579; PMID: 22186786; PMID: 22274398; PMID: 23966167; PMID: 24435915; PMID: 24924355; PMID: 25216526; PMID: 25402324; PMID: 25483077; PMID: 25528763; PMID: 25575815; PMID: 25714030; PMID: 25957417; PMID: 26047477; PMID: 26237416; PMID: 26375054; PMID: 26408681; PMID: 26431498; PMID: 26497690; PMID: 26859573; PMID: 27013582; PMID: 27105519; PMID: 27145267; PMID: 27152859; PMID: 27500926; PMID: 27683127; PMID: 27835903; PMID: 28030831; PMID: 28106277; PMID: 28296559; PMID: 28494180; PMID: 28622068; PMID: 28628234; PMID: 28903369; PMID: 29277768; PMID: 29374999; PMID: 29481803; PMID: 29932244; PMID: 29963961; PMID: 30003151; PMID: 30166061; PMID: 30292411; PMID: 30309504; PMID: 30673664; PMID: 30953192; PMID: 31208120; PMID: 31896528; PMID: 32366396; PMID: 34697138; PMID: 39477429; PMID: 39740880; PMID: 39993809; PMID: 40010967).

The potential lack of novelty of the manuscript overwhelms all other questions. The authors need to address this issue by comparing their results to similar published results on other tumor targeting bacteria. Their experiments are well performed, but this is not the main point.

Author Response

We thank the reviewer for highlighting the significant body of work on tumor-targeting bacteria, particularly the innovative studies from the Hoffman group involving genetically engineered Salmonella. These vectors, designed to colonize hypoxic tumor regions and stimulate antitumor immunity in murine models, represent a well-established therapeutic strategy.

Our study, however, focuses on a fundamentally different context. Staphylococcus aureus is a clinically significant opportunistic pathogen—not a therapeutic agent—that has been observed and cultured from breast cancer tissues (e.g., PMID: 40005832, 31025880). In contrast, Salmonella is not typically found in breast cancer and is not part of the endogenous tumor microbiota. While Salmonella is introduced exogenously in experimental models, S. aureus may naturally invade and persist within tumor cells, influencing tumor behavior and immune responses.

To our knowledge, this is the first systematic analysis of intracellular S. aureus persistence in human TNBC cell lines. We demonstrate cell line-dependent differences in bacterial clearance, toxin sensitivity, and growth suppression. These findings suggest a novel, non-therapeutic mechanism of tumor modulation, distinct from engineered bacterial therapies.

Thus, our work contributes a complementary perspective by examining the potential role of endogenous bacterial pathogens in breast cancer biology. This highlights a clinically relevant dimension of tumor–microbiome interaction, particularly under conditions of immune dysregulation or antibiotic exposure.

Reviewer 2 Report

Comments and Suggestions for Authors

The submitted article entitled “Cell Line-Dependent Internalization, Persistence, and Immunomodulatory Effects of Staphylococcus aureus in Triple-Negative Breast Cancer” by Kianpour Rad et al. reports the role of S. aureus in influencing the immune milieu of TNBC cells that have a potential for improving the therapeutic efficacy. To establish the above hypothesis, authors have used 3 TNBC cell lines (MDA-MB-231, MDA-MB-468, MDA-MB-453) where they have shown the increase of IFN g- induced PDL1 in S. aureus infected cells. They also tried to elucidate the mechanism of PDL1 increase related to phosphorylation of STAT. Finally, the authors tried to study the expression of IFN g-induced PDL1 in relation to TLR2 with Lipoteichoic acid in TNBC cells.

General Critique:

While your invitro study is novel and impressive, there are some experimental validations that need to be clarified and confirmed before considering it for publication.

Specific Critique:

  1. The authors tried to infer that different breast cancer cell lines exhibit varying capacity to internalize S. aureus by using 3 TNBC cell lines. The authors should support the above inference with 3 additional TNBC cell lines (SUM 159, BT-549, Hs578t, BT-20 or MB-436). Is there any particular logic behind using these 3 TNBC cells? The authors have performed FLOW to establish internalization. However, Figure S1C, Figure S1D and Figure S1E should be included in main Figure 1 and the present Figure 1C and Figure 1D can be moved to the supplementary. This will establish that the authors have tried various methods to establish the internalization of S. aureus in TNBC cell lines. Every experiment does not need to be done with 6 different TNBC cell lines. But different experiments including different types of TNBC cells will confirm the above inference by author. Scale bar and quantification graph of three independent experiments are required for Figures like S1C involving microscopy.
  2. The authors have studied the persistence by S. aureus in TNBC cell lines by quantifying colony forming units. It’s not clear about the significance statistics compared within the cell lines and should be performed in other TNBC cell lines too. It would be interesting to show the survivability status of these TNBC during the persistence of the S. aureus in these cell lines.
  3. The authors have used crystal violet and assays to confirm proliferation in Figure 4. It’s not clear what proliferation assay the authors have used. MTS assay can be used as a standard proliferation assay or Annexin V FLOW.
  4. The authors have established that S. aureus infection enhances IFN-Gamma induced PDL1 in a breast cell line dependent manner. The schema sub figure (Figure 5D) should be the first sub figure. It would be great if the author can also show the viability status of these TNBC cell lines (both short term as well as long term infected) in supplementary information.
  5. In Figure 6 the authors have tried to confirm the Interferon Gamma induced PDL1 with infection of the bacteria involves phosphorylation of STAT1. However, to infer this it is essential to perform a knockdown of STAT 1 and observe the PDL1 expression with and without infection as well as IFN gamma (at least in 1TNBC cell line).
  6. The blots in 6a (MDA-MB-231, MCF-12A) for Beta actin and STAT1 need to be repeated with a proper representative image.
  7. Schematic illustration of TLR2 experiment with/without infection (LTA) is very much required to understand the author’s hypothesis more precisely. The graph in Figure 7C should be improved for accurate depiction.

Minor Critique:

  1. In section 3.1 Figure 1A is missing.
  2. In section 3.6 the figure numbers in the text are incorrectly represented. Eg. Figure 7C represents PDL1 expression and not Figure 7B.
  3. Supplementary Figure S6, representative western blot, should be given with proper labelling. If asked by the journal, the author can give the raw images.

Author Response

The submitted article entitled “Cell Line-Dependent Internalization, Persistence, and Immunomodulatory Effects of Staphylococcus aureus in Triple-Negative Breast Cancer” by Kianpour Rad et al. reports the role of S. aureus in influencing the immune milieu of TNBC cells that have a potential for improving the therapeutic efficacy. To establish the above hypothesis, authors have used 3 TNBC cell lines (MDA-MB-231, MDA-MB-468, MDA-MB-453) where they have shown the increase of IFN g- induced PDL1 in S. aureus infected cells. They also tried to elucidate the mechanism of PDL1 increase related to phosphorylation of STAT. Finally, the authors tried to study the expression of IFN g-induced PDL1 in relation to TLR2 with Lipoteichoic acid in TNBC cells.

General Critique:

While your invitro study is novel and impressive, there are some experimental validations that need to be clarified and confirmed before considering it for publication.

Response:

We sincerely thank the reviewer for their thoughtful assessment and positive remarks regarding the novelty and overall quality of our study. We appreciate the acknowledgment of our work exploring the role of Staphylococcus aureus in modulating the immune response in triple-negative breast cancer (TNBC), and its potential implications for therapeutic efficacy.

We have carefully addressed all specific concerns and suggestions in the subsequent responses and revised the manuscript accordingly to clarify our experimental design, results, and interpretations. We believe these revisions have strengthened the clarity and scientific rigor of our study, and we are grateful for the reviewer’s constructive feedback, which has contributed meaningfully to the improvement of the manuscript.

Specific Critique:

2.1 The authors tried to infer that different breast cancer cell lines exhibit varying capacity to internalize S. aureus by using 3 TNBC cell lines. The authors should support the above inference with 3 additional TNBC cell lines (SUM 159, BT-549, Hs578t, BT-20 or MB-436). Is there any particular logic behind using these 3 TNBC cells?

Response:

We thank the reviewer for this valuable suggestion. While we agree that including additional TNBC cell lines such as SUM159, BT-549, Hs578T, BT-20, or MDA-MB-436 (all classified as claudin-low) would further strengthen our conclusions, constraints related to time and cell line availability prevented us from doing so in the current study. Nevertheless, we selected MDA-MB-231, MDA-MB-468, and MDA-MB-453 to represent distinct molecular subtypes of TNBC—namely, mesenchymal-like (MDA-MB-231), basal-like (MDA-MB-468), and HER2-enriched/TNBC variant (MDA-MB-453)—to capture a broad spectrum of TNBC heterogeneity. We have clarified this rationale in the manuscript and have now explicitly acknowledged the limitation in the Discussion, noting that future studies will aim to validate our findings using additional claudin-low TNBC models.

2.2 The authors have performed FLOW to establish internalization. However, Figure S1C, Figure S1D and Figure S1E should be included in main Figure 1 and the present Figure 1C and Figure 1D can be moved to the supplementary. This will establish that the authors have tried various methods to establish the internalization of S. aureus in TNBC cell lines.

Response:

We thank the reviewer for the thoughtful suggestion regarding figure placement. However, we believe the current figure organization offers a clearer and more cohesive presentation of our findings.

Figure 2 and Figure S1d present detailed and representative transmission electron microscopy (TEM) images, which directly visualize the intracellular localization and ultrastructural features of S. aureus within TNBC cells. Given that TEM is considered the gold standard for confirming bacterial internalization, we consider these data to be most appropriate for inclusion in the main figures.

Figure S1c utilizes an alternative method—fluorescence microscopy of CFSE-labeled bacteria combined with propidium iodide (PI) staining—to differentiate extracellular from intracellular bacteria. While this technique further supports our internalization findings, it uses a distinct labeling strategy and serves as a complementary validation rather than a primary demonstration.

Additionally, Figure S1e confirms the efficacy of the gentamicin protection assay, providing functional evidence for the presence of viable intracellular bacteria.

Since the main text already integrates multiple orthogonal approaches—including flow cytometry, gentamicin protection, and TEM—we believe the current layout effectively communicates our key findings without overloading the main figures. Therefore, we respectfully propose retaining the existing figure arrangement.

2.3 Every experiment does not need to be done with 6 different TNBC cell lines. But different experiments including different types of TNBC cells will confirm the above inference by author.

Response:

We respectfully note that this comment has been addressed earlier (response 2.1). While it is not necessary to perform every experiment using six different TNBC cell lines, we have included multiple TNBC cell lines across different experiments to support and confirm our conclusions.

2.4 Scale bar and quantification graph of three independent experiments are required for Figures like S1C involving microscopy.

Response:

We thank the reviewer for this comment. Figure S1c was intended as a qualitative confirmation of the effective elimination of viable extracellular S. aureus following gentamicin treatment, using propidium iodide (PI) to selectively stain non-viable extracellular bacteria. As described in the figure legend (Figure S1c), CFSE-labeled S. aureus that remained intracellular retained green fluorescence, while extracellular bacteria were stained red with PI. Given the qualitative nature and purpose of this assay—to confirm the efficacy of the antibiotic treatment step—quantification of the images was not performed, as it would not provide additional meaningful data. However, we have now included scale bars in the figure to improve clarity and reproducibility, as requested.

2.5 The authors have studied the persistence by S. aureus in TNBC cell lines by quantifying colony forming units. It’s not clear about the significance statistics compared within the cell lines and should be performed in other TNBC cell lines too. It would be interesting to show the survivability status of these TNBC during the persistence of the S. aureus in these cell lines.

Response:

We thank the reviewer for this constructive feedback. To clarify, statistical comparisons of bacterial clearance between cell lines were performed by comparing the slopes of linear regression models fitted to CFU counts over time, using two-tailed t-tests. These comparisons are now explicitly described in the revised legend for Figure 2, and the 95% confidence intervals for each slope estimate are included in the Results to highlight differences in bacterial clearance kinetics. While individual p-values were not provided, these confidence intervals offer a clear indication of statistical significance.

Regarding the expansion to additional TNBC cell lines, we have addressed this point in our response to Comment 2.1.

Lastly, we appreciate the reviewer’s interest in the viability status of TNBC cells during persistent infection. This aspect is addressed in Section 3.4, where we detail the cell line–dependent cytotoxic effects of S. aureus and the recovery of cell viability over time, indicating how bacterial persistence differentially impacts host cell survival and proliferation.

2.6 The authors have used crystal violet and assays to confirm proliferation in Figure 4. It’s not clear what proliferation assay the authors have used. MTS assay can be used as a standard proliferation assay or Annexin V FLOW.

Response:

We thank the reviewer for this valuable comment. As clarified in the Results section (3.4), the long-term effects of S. aureus infection on cell proliferation were assessed using a crystal violet assay over five days. This colorimetric assay quantifies total biomass and is commonly used as a reliable surrogate for cumulative cell proliferation over time in adherent cell lines. We acknowledge that other measures of metabolic activity (e.g., MTS assay) or apoptosis (e.g., Annexin V flow cytometry) also could provide complementary insights. However, our primary aim was to evaluate sustained growth potential following infection, for which the crystal violet assay is well suited. To improve clarity, we have revised the text in the Results section to explicitly state that a crystal violet proliferation assay was used and have added a brief justification for its use.

2.7 The authors have established that S. aureus infection enhances IFN-Gamma induced PDL1 in a breast cell line dependent manner. The schema sub figure (Figure 5D) should be the first sub figure.

Response:

We thank the reviewer for this constructive suggestion. We agree that presenting the schematic overview first improves the logical flow and clarifies the experimental design. Accordingly, we have reordered the figure panels so that the schema now appears as Figure 5a, with all subsequent panels re-labeled in the figure, legend, and main text. The figure legend has also been updated to reflect this change.

2.8 It would be great if the author can also show the viability status of these TNBC cell lines (both short term as well as long term infected) in supplementary information.

Response:

We thank the reviewer for this helpful comment. Data on TNBC cell viability following both short-term and long-term S. aureus infection are presented in Figure 4a–b, as we consider them critical for interpreting the impact of bacterial persistence. However, we have not investigated the effects of interferon-gamma (IFN-γ) treatment on cell viability or proliferation in this context. We agree that such data could provide valuable insight into how S. aureus-infected TNBC cells respond to inflammatory stimuli, particularly in relation to immune-mediated cytotoxicity or potential resistance mechanisms, and this represents an interesting avenue for future investigation.

2.9 In Figure 6 the authors have tried to confirm the Interferon Gamma induced PDL1 with infection of the bacteria involves phosphorylation of STAT1. However, to infer this it is essential to perform a knockdown of STAT 1 and observe the PDL1 expression with and without infection as well as IFN gamma (at least in 1TNBC cell line).

Response:

We thank the reviewer for this insightful suggestion. It is well established that IFN-γ induces PD-L1 expression through the JAK–STAT1 signaling pathway, and this mechanism has been demonstrated across numerous cancer types, including breast cancer. In our study, we observed that S. aureus infection enhances IFN-γ–induced PD-L1 expression in a cell line–dependent manner, and this is accompanied by increased phosphorylation of STAT1. These findings are consistent with the known IFN-γ–STAT1–PD-L1 signaling axis and support a role for STAT1 activation in mediating the observed response.

We agree that STAT1 knockdown would provide direct mechanistic confirmation of its necessity in this context, particularly to determine whether the infection-enhanced PD-L1 expression is STAT1-dependent. However, due to technical and resource constraints, we are currently unable to perform these additional experiments. We have acknowledged this limitation in the revised Discussion section and proposed it as an important avenue for future mechanistic investigation.

2.10 The blots in 6a (MDA-MB-231, MCF-12A) for Beta actin and STAT1 need to be repeated with a proper representative image.

Response:

We thank the reviewer for this helpful comment. In response, we have replaced the representative blots for β-actin and STAT1 in MDA-MB-231 and MCF-12A cells in Figure 6a with clearer images from replicate experiments to enhance visual clarity while preserving the integrity of the original findings. Each western blot experiment was independently performed 2 to 3 times, yielding consistent results across replicates. As per journal guidelines, we have renumbered the uncropped western blots for all data presented in Figure 6 in the Supplementary Material (Fig. S6 -9).

2.11 Schematic illustration of TLR2 experiment with/without infection (LTA) is very much required to understand the author’s hypothesis more precisely. The graph in Figure 7C should be improved for accurate depiction.

Response:

We thank the reviewer for the suggestion to include a schematic illustration of the TLR2 experiment to clarify the experimental design (Figure 7). While we appreciate the value of such visual summaries, we believe the revised figure layout and improved graph labeling now sufficiently convey the experimental conditions and findings. Specifically, we have redrawn the graphs in Figure 7c to enhance clarity and have added explicit treatment labels for each condition to ensure accurate interpretation. We have also revised the figure legend for added clarity. We hope these updates adequately address the reviewer’s concerns and improve the reader's understanding of the TLR2 experiments.

Minor Critique:

2.13 In section 3.1 Figure 1A is missing.

Response:

Thank you for pointing this out. We have corrected this issue.

2.14 In section 3.6 the figure numbers in the text are incorrectly represented. Eg. Figure 7C represents PDL1 expression and not Figure 7B.

Response:

Thank you for pointing this out. We have corrected this issue.

2.15 Supplementary Figure S6, representative western blot, should be given with proper labelling. If asked by the journal, the author can give the raw images.

Response:

Thank you for your suggestion. We have revised the Supplementary Material to include all uncropped western blot images with appropriate labeling, in accordance with the journal’s guidelines (Fig. S6–S9).

Reviewer 3 Report

Comments and Suggestions for Authors

The authors investigated the impact of Staphylococcus aureus on triple-negative breast cancer (TNBC), focusing on bacterial internalization, persistence, and immunomodulatory effects. Notably, they demonstrated that TNBC cells can internalize S. aureus, leading to upregulation of PD-L1 expression, particularly when combined with IFN-γ treatment. The manuscript provides sufficient background and the experimental methods are described in adequate detail.

Recommendations:

  1. Figure 1B: The column titles contain multiple typographical errors. Please revise for clarity and accuracy.
  2. Figure 6A: The quality of the Western blot images, particularly for ACTB and total STAT1, is suboptimal. The authors should consider repeating the experiment or providing higher-quality images to support their findings more convincingly

Author Response

The authors investigated the impact of Staphylococcus aureus on triple-negative breast cancer (TNBC), focusing on bacterial internalization, persistence, and immunomodulatory effects. Notably, they demonstrated that TNBC cells can internalize S. aureus, leading to upregulation of PD-L1 expression, particularly when combined with IFN-γ treatment. The manuscript provides sufficient background and the experimental methods are described in adequate detail.

Response:

We thank the reviewer for their positive evaluation of our manuscript. We also appreciate the acknowledgment of the clarity and adequacy of our background information and methodological descriptions.

Recommendations:

  • Figure 1B: The column titles contain multiple typographical errors. Please revise for clarity and accuracy.

Response:

We thank the reviewer for this comment. We would like to clarify that the letters above the bars in Figure 1b and 1c are not column titles but denote statistical groupings based on Tukey’s multiple comparisons test, where different letters indicate statistically significant differences between cell lines (p < 0.0001). To avoid confusion, we have revised the figure legend to clearly state the meaning of these letter labels and confirm that no typographical errors are present in the column titles. The updated legend now reads:

“... Different letters denote statistically significant differences between cell lines (p < 0.0001). Full statistical details are provided in Table S1.”

We hope this revision clarifies the presentation and addresses the reviewer’s concern.

  • Figure 6A: The quality of the Western blot images, particularly for ACTB and total STAT1, is suboptimal. The authors should consider repeating the experiment or providing higher-quality images to support their findings more convincingly.

Response:

We thank the reviewer for their valuable feedback. We have modified Figure 6a, by providing higher-quality images. We wish to emphasize that the data shown in Figure 6a were representative examples of experiments independently repeated at least twice, with similar results obtained in each case (uncropped data presented in Supplementary Material (Fig. S6-9). Moreover, the p-STAT1 blots—which are central to our conclusions—are of high quality and clearly demonstrate the expected modulation in response to S. aureus and IFN-γ. We ensured that the uncropped blots presented in Supplementary Material Fig. S6-9 are clearly labelled, and that the densitometry for all data are provided for each blot.

Round 2

Reviewer 1 Report

Comments and Suggestions for Authors The authors are non-responsive to the novelty criticism.  

Author Response

Comment 1: The authors are non-responsive to the novelty criticism.

Response 1: We thank the reviewer for highlighting the importance of clarifying the novelty of our findings. In response, we have extensively revised the Introduction (paragraphs 3–8) to more explicitly distinguish our study from previous work and emphasize its unique contributions.

While Liu et al. (2024) reported that Staphylococcus aureus supernatant and α-hemolysin enhanced PD-L1 expression and sensitivity to immune checkpoint blockade in murine TNBC models (4T1 and EO771) in vivo, our study provides several novel insights:

Use of Well-Characterised, Molecularly Distinct Human Breast Cell Lines
Our study utilized three well-characterized, molecularly distinct human TNBC cell lines—MDA-MB-231, MDA-MB-453, and MDA-MB-468—alongside the non-tumorigenic MCF-12A breast epithelial line. Compared to murine TNBC models, these human cell lines provide greater clinical relevance, particularly for studying PD-L1–targeted immunotherapeutic pathways. Notably, we observed that PD-L1 upregulation in response to S. aureus was cell line–dependent, reflecting subtype-specific variation in bacterial responsiveness. This heterogeneity—absent in previous studies using murine TNBC cells—underscores a novel and clinically important finding: not all TNBC subtypes increase PD-L1 expression in response to S. aureus or its components.

Endogenous vs. Exogenous Bacteria
Whereas Liu et al. and others (including the Hoffman lab) have focused on the therapeutic delivery of exogenous bacteria and/or bacterial products, our study investigates the functional consequences of viable endogenous S. aureus that persist within human TNBC cells—a scenario that more closely reflects intratumoral microbial colonization observed in patient tumors. To our knowledge, this is the first study to directly demonstrate that viable intracellular S. aureus can modulate PD-L1 expression in human TNBC cells.

TLR2-Dependent Potentiation of IFN-γ Signaling
To our knowledge, this is the first study to demonstrate that TLR2 agonists—lipoteichoic acid (LTA) and Pam3CSK4—can enhance IFN-γ–induced PD-L1 expression in human TNBC cell lines. While prior studies have shown that Staphylococcus aureus can activate TLR2 signaling and upregulate PD-L1 expression in immune cells (eg. Wang et al., 2012; PMID: 22930672) and head and neck cancer cell lines (Mann et al., 2021; PMID: 34638266), our findings uncover a previously unreported synergistic interaction in TNBC. Specifically, our data suggest that bacterial TLR2 ligands can amplify IFN-γ–driven immune checkpoint signaling, providing new insight into how endogenous microbial components may contribute to immune evasion in the TNBC tumor microenvironment.

Controlled In Vitro Mechanistic Investigation
Although our study was conducted in vitro, this design enabled rigorous control over experimental variables, such as bacterial dose, viability, and host cell context—factors that are inherently difficult to isolate in vivo. This approach allowed us to dissect cell line–specific interactions and establish mechanistic links between bacterial infection, TLR2 signaling, and PD-L1 regulation.

Taken together, our findings identify a previously unrecognized role for viable intracellular S. aureus and TLR2 signaling in modulating immune checkpoint expression in human TNBC cells. This mechanistic insight addresses a significant gap in the literature and underscores the therapeutic potential of targeting host–microbe interactions to enhance immunotherapy responsiveness in breast cancer.

Further to our Round 1 response, we would like to re-emphasise why we focused on Staphylococcus rather than Salmonella, as suggested by the reviewer. Please note that the references provided here represent only a subset; most supporting citations can be found within the manuscript.

This study, like many others (PMID: 38191648, 35407638), focuses specifically on bacteria that are endogenously present within the breast microenvironment. Therefore, we did not conduct any experiments on microbes that are not typically found in this niche. The overarching question guiding our study is: What impact do these endogenous microbes have on cancer cells? Accordingly, we selected Staphylococcus as our focus, since both our systematic review and metaanalysis (PMID: 40005832) and previous culture-based studies (PMID: 31025880) have shown that Staphylococcus is commonly found and abundant within the breast microenvironment.

For example, PMID: 31025880 performed a meta-analysis combining sequencing and culture methods to identify bacteria present in breast milk, mastitis, normal, and abscess tissue samples. Salmonella enterica was the only Salmonella species cultured, and it appeared in just 16 out of 216 studies (frequency: 7%), predominantly in infection-related contexts such as mastitis or abscesses. In contrast, 21 Staphylococcusspecies were detected in breast-related studies (frequency: ~69%), and even when considering only Staphylococcus aureus, 58 out of 216 studies (frequency: 26%) successfully cultured the species (PMID: 31025880). Combined with our sequencing data (PMID: 40005832), these findings strengthen our confidence that Staphylococcus aureus is indeed present within the breast microenvironment.

Furthermore, the mechanisms by which these bacteria enter host cells differ fundamentally. S. aureus uses a zipper mechanism, mediated by fibronectin-binding proteins that link to host α5β1 integrins, resulting in receptor-mediated uptake (PMID: 17021255, 11207545). In contrast, Salmonella employs a trigger mechanism involving a Type III Secretion System, which injects effector proteins to induce cytoskeletal rearrangements and membrane ruffling (PMID: 1360615, 11687484, 18026123). These distinct biological strategies further highlight the mechanistic divergence between S. aureus and therapeutic Salmonella strains and support our decision not to cite this literature.

Thus, our use of Staphylococcus—as opposed to Salmonella, which is rarely found in the breast cancer setting—is well-justified and consistent with our study’s aim to explore the influence of endogenous microbes on cancer biology.

Reviewer 2 Report

Comments and Suggestions for Authors

The submitted article entitled “Cell Line-Dependent Internalization, Persistence, and Immunomodulatory Effects of Staphylococcus aureus in Triple-Negative Breast Cancer” by Kianpour Rad et al. reports the role of S. aureus in influencing the immune milieu of TNBC cells that have a potential for improving the therapeutic efficacy. To establish the above hypothesis, authors have used 3 TNBC cell lines (MDA-MB-231, MDA-MB-468, MDA-MB-453) where they have shown the increase of IFN g- induced PDL1 in S. aureus infected cells. They also tried to elucidate the mechanism of PDL1 increase related to phosphorylation of STAT. Finally, the authors tried to study the expression of IFN g-induced PDL1 in relation to TLR2 with Lipoteichoic acid in TNBC cells.

Major Critique:

  1. The author has made response against the requirement of proving the essentiality of the main hypothesis based on 3 different categories of TNBC- namely, mesenchymal-like (MDA-MB-231), basal-like (MDA-MB-468), and HER2-enriched/TNBC variant (MDA-MB-453)—to capture a broad spectrum of TNBC heterogeneity. The authors have also mentioned that due to constrain of time and due to claudin low TNBC group the other TNBC cells suggested by the reviewer (SUM159, BT-549, Hs578T, BT-20, or MDA-MB-436 lines) are not used. However, there is no experimental proof in the manuscript showing the relationship between claudin low group of TNBC cells and internalization efficacy of Staphylococcus aureus. Does the author show any negative relationship between the claudin low group of TNBCS and the internalization /immunomodulatory effect by S. aureus in supplementary data?

If not, then based on only 3 cell lines the title doesn’t fit for   all TNBC. Alternatively, authors can prove this hypothesis either in an in vivo model, using MDA-MB-231 cell or 3D spheroid model using these 3 cell lines.

  1. To establish the mechanistic role of STAT 1, KD by at least si RNA strategy is essential to confirm the hypothesis.
  2. Figure 7, instead of IFN-y it should be IFN-ɣ

Author Response

Comment 1: The author has made response against the requirement of proving the essentiality of the main hypothesis based on 3 different categories of TNBC- namely, mesenchymal-like (MDA-MB-231), basal-like (MDA-MB-468), and HER2-enriched/TNBC variant (MDA-MB-453)—to capture a broad spectrum of TNBC heterogeneity. The authors have also mentioned that due to constrain of time and due to claudin low TNBC group the other TNBC cells suggested by the reviewer (SUM159, BT-549, Hs578T, BT-20, or MDA-MB-436 lines) are not used. However, there is no experimental proof in the manuscript showing the relationship between claudin low group of TNBC cells and internalization efficacy of Staphylococcus aureus. Does the author show any negative relationship between the claudin low group of TNBCS and the internalization /immunomodulatory effect by S. aureus in supplementary data? If not, then based on only 3 cell lines the title doesn’t fit for   all TNBC. Alternatively, authors can prove this hypothesis either in an in vivo model, using MDA-MB-231 cell or 3D spheroid model using these 3 cell lines.

Response 1: We thank the reviewer for their thoughtful and constructive comment. We fully acknowledge the heterogeneity of triple-negative breast cancer (TNBC) and agree that no single study can completely capture the diversity of this subtype. Our rationale for selecting MDA-MB-231 (mesenchymal stem-like), MDA-MB-453 (LAR), and MDA-MB-468 (basal-like 1) was to represent three of the major molecular subtypes of TNBC as originally defined by Lehmann et al (PMID: 21633166; PMID: 27310713). Importantly, these subtypes reflect distinct transcriptomic profiles and therapeutic vulnerabilities, and collectively account for a substantial proportion of TNBC cases. For example, basal-like subtypes comprise the majority (~70–80%) of TNBC tumors, while mesenchymal and LAR subtypes each represent smaller but clinically relevant fractions (~10–15% each).

To better highlight this in the manuscript, we have further revised the opening sentence of the Results section as follows:

We investigated the internalization of S. aureus ATCC 25923 across triple-negative breast cancer (TNBC) cell lines representing the most prevalent molecular subtypes: MDA-MB-468 (basal-like 1), MDA-MB-231 (mesenchymal stem-like), and MDA-MB-453 (luminal androgen receptor), as well as the non-tumorigenic MCF-12A breast epithelial cell line

To our knowledge, this is the first study to directly test the effects of viable Staphylococcus aureus on a range of human TNBC cell lines representing distinct molecular subtypes. Previous studies have typically relied on murine TNBC models using S. aureus spent media or purified bacterial components (e.g., Liu et al., 2014, PMID: 38191648), and few have addressed the role of internalized live S. aureus in human TNBC. We believe this is a key strength of our study and contributes novel insights into how endogenous bacteria may modulate immune checkpoint signaling in a subtype-specific manner.

We recognize the reviewer’s interest in claudin-low TNBC cell lines such as SUM159, BT-549, and Hs578T, which are frequently cited as models of this subtype. Claudin-low breast cancers are estimated to make up approximately 7–14% of TNBCs (eg. PMID: 20813035; PMID: 25277734). While recent studies have promoted the claudin-low classification due to its association with features such as EMT, stemness, and immune suppression, it is important to note that this subtype remains controversial—particularly in the context of cell lines (PMID: 32647202). Transcriptomic instability and culture conditions can influence the expression of claudins and other defining markers, leading to ambiguity in classification (PMID: 20964822; PMID: 32647202). Some analyses have questioned whether claudin-low represents a true intrinsic subtype or a dynamic cell state, especially in vitro. Therefore, while these models are of biological interest, results derived from claudin-low cell lines should be interpreted with caution, and we believe that their use introduces complexities beyond the scope of the current study. Nevertheless, we acknowledge this as a potentially valuable direction for future research.

We did not include these additional cell lines in the current study due to resource constraints and scope, and we did not claim or aim to define a relationship between claudin-low status and S. aureus internalization or immunomodulatory activity. No such analysis is included in the manuscript or supplementary data.

Importantly, our manuscript explicitly presents the findings as cell line–dependent, as reflected in the title: “Cell Line-Dependent Internalization, Persistence, and Immunomodulatory Effects of Staphylococcus aureus in Triple-Negative Breast Cancer.” Rather than generalizing to all TNBC, we emphasize that the observed effects differ across subtypes, reinforcing the importance of molecular context.

We agree that future studies should extend this work by including additional TNBC models—such as claudin-low—and incorporating more complex systems such as 3D spheroids and in vivo models. These directions are part of our planned follow-up and have been acknowledged in the revised Discussion section:

Future studies should validate these findings under more physiologically relevant con-ditions by incorporating additional TNBC subtypes, including claudin-low models, as well as patient-derived organoids, 3D spheroids, or in vivo systems. Given the hetero-geneity of TNBC, such approaches will be essential to more comprehensively define the relevance of bacterial internalization and immunomodulatory effects across the broader TNBC landscape.”

We hope this revised explanation addresses the reviewer’s concerns and supports the scope and title of our study as appropriate for this initial investigation.

Comment 2: The authors should perform STAT1 knockdown experiments to confirm that the observed PD-L1 upregulation in the presence of S. aureus and IFN-γ is mediated by STAT1.

Response 2: We thank the reviewer for reiterating this important point. We fully agree that STAT1 is a critical transcriptional regulator of PD-L1 and that knockdown experiments would provide definitive evidence for its role in mediating the observed response.

However, the primary aim of our current study was to determine whether viable Staphylococcus aureus—an endogenous, facultative intracellular bacterium frequently detected in human breast tumors—can modulate immune checkpoint signaling in human TNBC cells. Our findings demonstrate that S. aureus selectively enhances IFN-γ–induced PD-L1 expression in human TNBC cell lines that internalize the bacteria (MDA-MB-231 and MDA-MB-468), and that this enhancement is accompanied by increased phosphorylation of STAT1, without a change in total STAT1 levels.

Crucially, this response is absent in MDA-MB-453 and MCF12A cells, which do not internalize S. aureus, reinforcing a functional link between bacterial uptake and potentiation of canonical IFN-γ/STAT1 signaling. We also show that TLR2 agonists (Pam3CSK4 and LTA) can similarly potentiate IFN-γ–induced PD-L1 expression—a novel observation, to our knowledge, in the context of human TNBC cells. Together, these data provide strong indirect evidence of a STAT1-dependent mechanism.

While we acknowledge that STAT1 knockdown would offer direct mechanistic validation, these experiments are technically demanding and outside the current scope due to resource constraints. Further, we wish to emphasise that the journal require a response to reviewers within 10 days. We are actively pursuing STAT1-targeted perturbations as part of our ongoing work and have now explicitly acknowledged this in the revised Discussion section:

Further, while our findings suggest that bacterial enhancement of PD-L1 expression is mediated through increased STAT1 phosphorylation, future studies incorporating STAT1 knockdown or pharmacologic inhibition will be essential to definitively establish causality.”

We respectfully submit that the current study provides a robust and novel dataset demonstrating a cell line–specific, infection-dependent potentiation of IFN-γ–STAT1–PD-L1 signaling, supported by Western blotting and flow cytometry. We believe these findings are sufficient to establish a compelling new framework for the role of endogenous bacteria in modulating immune checkpoint signaling in breast cancer, and we hope the reviewer finds this rationale acceptable.

Comment 3: Figure 7, instead of IFN-y it should be IFN-ɣ

Response 3:We thank the reviewer for pointing out this oversight. We have corrected the notation and ensured that the same Greek gamma symbol (g) is now used consistently throughout Figures 5, 6, and 7. We appreciate the reviewer’s attention to detail.

Reviewer 3 Report

Comments and Suggestions for Authors

I find the authors’ responses to be satisfactory and adequately address the concerns raised.

Author Response

Comments 1: I find the authors’ responses to be satisfactory and adequately address the concerns raised.

Response 1: We sincerely thank the reviewer for their thoughtful evaluation and positive feedback. We appreciate your time and constructive comments, which have helped strengthen our manuscript.

Round 3

Reviewer 2 Report

Comments and Suggestions for Authors

 The author has made response against the requirement of proving the essentiality of the main hypothesis based on 3 different categories of TNBC- namely, mesenchymal-like (MDA-MB-231), basal-like (MDA-MB-468), and HER2-enriched/TNBC variant (MDA-MB-453)—to capture a broad spectrum of TNBC heterogeneity. However, solely based on 3 different TNBC cell line, invitro data it cannot be concluded, as reflected in the title: “Cell Line-Dependent Internalization, Persistence, and Immunomodulatory Effects of Staphylococcus aureus in Triple-Negative Breast Cancer.”

Hence, this manuscript needs further validations and experiments to prove the proposed title. Without its validation in increased number of cell lines the article loses its novel hypothesis.